# Deep neural network trained on gigapixel images improves lymph node metastasis detection in clinical settings

Shih-Chiang Huang [1,2,9], Chi-Chung Chen [3,9], Jui Lan[4], Tsan-Yu Hsieh[5], Huei-Chieh Chuang[6], Meng-Yao Chien [3], Tao-Sheng Ou [3], Kuang-Hua Chen[1], Ren-Chin Wu [1], Yu-Jen Liu[1], Chi-Tung Cheng[7], Yu-Jen Huang[8], Liang-Wei Tao [8], An-Fong Hwu[8], I-Chieh Lin[1], Shih-Hao Hung[8], Chao-Yuan Yeh [3✉] & Tse-Ching Chen [1✉]

The pathological identification of lymph node (LN) metastasis is demanding and tedious. Although convolutional neural networks (CNNs) possess considerable potential in improving the process, the ultrahigh-resolution of whole slide images hinders the development of a clinically applicable solution. We design an artificial-intelligence-assisted LN assessment workflow to facilitate the routine counting of metastatic LNs. Unlike previous patch-based approaches, our proposed method trains CNNs by using 5-gigapixel images, obviating the need for lesion-level annotations. Trained on 5907 LN images, our algorithm identifies metastatic LNs in gastric cancer with a slide-level area under the receiver operating characteristic curve (AUC) of 0.9936. Clinical experiments reveal that the workflow significantly improves the sensitivity of micrometastasis identification (81.94% to 95.83%, $P < .001$) and isolated tumor cells (67.95% to 96.15%, $P < .001$) in a significantly shorter review time (−31.5%, $P < .001$). Cross-site evaluation indicates that the algorithm is highly robust (AUC = 0.9829).

[1] Department of Anatomic Pathology, Linkou Chang Gung Memorial Hospital, Chang Gung University, College of Medicine, Taoyuan, Taiwan. [2] Graduate Institute of Clinical Medical Sciences, College of Medicine, Chang Gung University, Taoyuan, Taiwan. [3] aetherAI Co., Ltd., Taipei, Taiwan. [4] Department of Anatomic Pathology, Kaohsiung Chang Gung Memorial Hospital, Chang Gung University, College of Medicine, Kaohsiung, Taiwan. [5] Department of Anatomic Pathology, Keelung Chang Gung Memorial Hospital, Chang Gung University, College of Medicine, Keelung, Taiwan. [6] Department of Anatomic Pathology, Chiayi Chang Gung Memorial Hospital, Chang Gung University, College of Medicine, Chiayi, Taiwan. [7] Department of Surgery, Linkou Chang Gung Memorial Hospital, Chang Gung University, College of Medicine, Taoyuan, Taiwan. [8] Department of Computer Science and Information Engineering, National Taiwan University, Taipei, Taiwan. [9] These authors contributed equally: Shih-Chiang Huang, Chi-Chung Chen. ✉email: joeyeh@aetherai.com; ctc323@cgmh.org.tw

Globally, gastric carcinoma is the fifth most common cancer and the fourth leading cause of cancer-related mortality[1]. In 2020, the estimated number of new cases exceeded 1 million, with the highest incidence being in Eastern Asia, Eastern Europe, and Latin America[1]. In East Asia, the average incidence of gastric cancer is 32.5 and 13.2 per 100,000 people among men and women, respectively. Accurate staging is crucial for the proper treatment of patients with gastric cancer. The American Joint Committee on Cancer (AJCC) staging system, the most commonly used staging system, involves the assessment of parameters T, N, and M. These refer to the extent of the primary tumor, the involvement of regional lymph nodes (LNs), and distant metastasis, respectively[2]. The gold standard for the diagnosis of gastric cancer is the histologic evaluation of surgical specimens, through which accurate pathological staging results can be obtained. The challenge of identifying metastatic carcinoma in LNs in gastric cancer is attributable to the high percentage of diffuse and mixed-type cancer, which accounts for more than 40% of cases[3]. In diffuse and mixed-type gastric cancer, metastatic tumor cells may be poorly cohesive small clusters or individual cells. They may resemble histiocytes or lymphocytes in appearance, having no well-adherent aggregates, glandular structures, and easily recognizable nuclear pleomorphism. Although the prognostic impact of detecting micrometastases[4–6] (≥0.2 mm, <2 mm) and isolated tumor cells[5,7–9] (ITCs; <0.2 mm) remains under debate, promptly detecting metastasis is recommended to provide more information for clinical treatment[10]. The already demanding identification of small metastases in LNs becomes even more challenging when numerous harvested LNs are involved. The numbers of LNs harvested from radical gastrectomy are typically high. In 186 consecutive specimens collected at Linkou Chang Gung Memorial Hospital (CGMH), the average number of LNs retrieved from this procedure was approximately 38 (7186/186). The AJCC staging system requires a minimum of 15 LNs to be harvested. The fact that LN metastasis identification is demanding and tedious makes it suitable for the application of an artificial intelligence (AI)-assisted workflow.

Since 2012, convolutional neural networks (CNNs) have facilitated advances in deep learning with regard to image recognition and classification tasks. Cutting-edge AI technology has permeated medical imaging and computational pathology[11–13]. Studies have demonstrated that the ability of CNNs to detect metastatic LNs in gastric cancer cases is comparable to that of human experts[14,15]. However, these algorithms have not been incorporated into workflows for real-world application in clinical pathology. The main reason is that whole slide images (WSIs) have an ultrahigh spatial resolution, typically in the range of billions of pixels, making them extremely large and highly difficult to process under a typical CNN training workflow. This problem can be circumvented through two methods. The first is the strongly supervised patch-based approach[14–16], in which WSIs are divided into small patches and the labels for each image patch are derived from detailed annotations made by pathologists. The efforts and complexity of annotation involved in this process are prohibitively high; thus, the availability of annotated data is limited, hindering the accuracy improvement of CNNs benefited from the gain of data. Most previous works[11,14,15] for LN metastasis identification adopted this method. Therefore, this study focused on the second approach, weakly supervised learning, which does not entail detailed annotation and allows annotations to be in the form of positive or negative LNs. However, some weakly supervised methods adhere to a two-stage workflow (e.g., multiple instance learning [MIL]), the performance of which is only comparable to that of the patch-based method when numerous WSIs are used for training[16–21]. Other methods, which employ end-to-end single-stage training, are only feasible for tasks performed under low magnification (up to 23,000 × 23,000 pixels, 4× magnification [2.3 μm/pixel])[22–25]. In view of these drawbacks, we developed an end-to-end weakly supervised method called enhanced streaming CNN (ESCNN). It substantially boosted the throughput and reduced the memory requirement by moving image augmentation into the streaming CNN[24] pipeline and skipping unnecessary computations. These improvements allowed an increase of input image resolution to 75,000 × 75,000 pixels (20× magnification, 0.46 μm/pixel), enabling the direct training of gigapixel images for metastasis identification.

To ensure its practicality, we further combined the model with a pathological LN assessment workflow. Accordingly, an AI-assisted LN assessment workflow was developed in a real-world setting. The workflow comprised two modules: an LN detector module for counting LNs and a metastasis identification module. Upon the import of corresponding WSIs, the workflow was triggered immediately, and then prediagnostic results were presented for pathologists to conduct final assessments. To determine the effectiveness of the workflow, we designed a clinical experiment simulating a pathologist's routine LN assessment procedure. In the experiments, six pathologists reviewed 80 slides with and without AI assistance, and the review time and number of LNs positive and negative for metastatic carcinoma were recorded.

## Results

**AI-assisted LN assessment workflow**. The clinical workflow interface is displayed in Fig. 1. Upon the import of a WSI, the LN detector is triggered to outline the LNs, after which the LN metastasis identification module classifies each LN as positive or negative and highlights the lesion area. To address false predictions, pathologists can edit contours, contour labels, or amend the final counts for correction. To assist pathologists with N-category assessment, a panel summarizing the numbers of positive and negative LNs of the current slide and study was employed. The evaluation of the ESCNN is presented as follows, followed by a discussion of the proposed weakly supervised end-to-end training method for metastasis identification and the clinical evaluation of the AI-assisted workflow. A demo video is provided as Supplementary Movie 1.

**ESCNN performance in metastasis identification**. The experiments were conducted using the main training set, which consisted of 983 WSIs including 5907 LN images collected from Linkou CGMH in 2019. Each LN image was downscaled to 20× magnification (0.46 μm/pixel) and padded to dimensions of 75,000 × 75,000 pixels. The metastasis identification model of the ResNet50 architecture[26] was trained using the ESCNN in an end-to-end, weakly supervised manner. The model was then tested using the main test set of 1156 LN images (positive: 295; negative: 861) collected by Linkou CGMH in 2019. The ground truth of each LN image was reviewed by four pathologists (S.-C.H., J.L., H.-C.C., and T.-Y.H.) and meticulously examined by the most experienced pathologist (S.-C.H., an expert in gastric cancer pathology) with the assistance of immunohistochemistry (IHC) testing. The model achieved an area under the receiver operating characteristic curve (AUC) of 0.9831 (0.9728–0.9934) for the classification of LN images. After the LN prediction scores were aggregated according to their maxima, the slide-level AUC reached 0.9936 (0.9856–1.0000), comparable to the slide-level AUC of 0.986 of a patch-based model trained with 700 fully annotated WSIs[15]. Thus, the effectiveness of weak supervision with less annotation effort was demonstrated. To investigate the

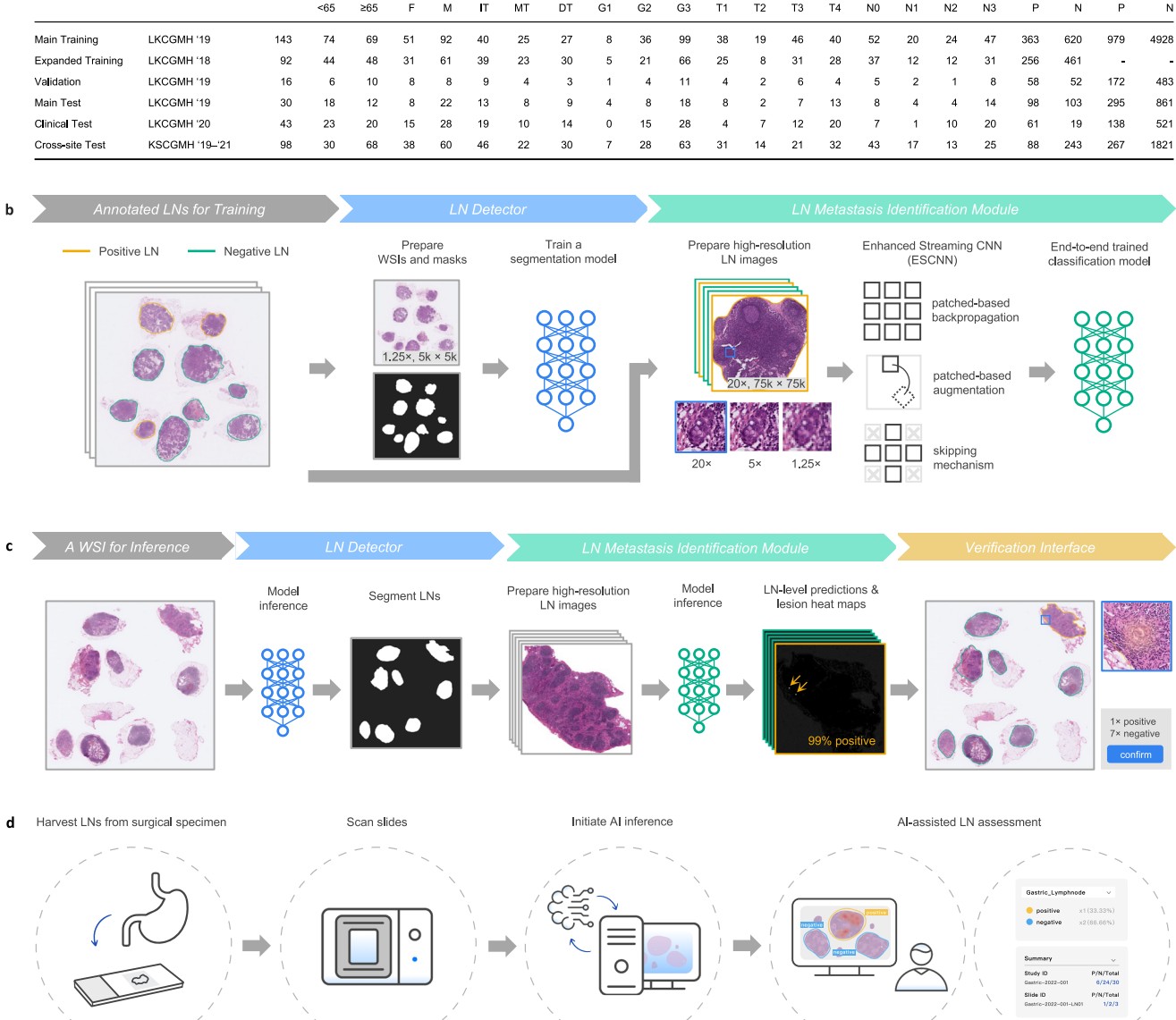

**Fig. 1 Overview of the gastric LN assessment workflow. a** Description of the data sets in this study, including the number (P: positive, N: negative) of studies, slides, and LNs; the distributions of age, sex (F: female, M: male), and the Lauren classification (IT: intestinal type, MT: mixed type, DT: diffuse type), the grading of gastric cancer, and AJCC T and N categories. **b** Pipeline for training the LN detector and the metastasis identification module. **c** The inference pipeline leverages the trained models to provide prediagnostic predictions of positive and negative LN counts and highlights suspicious areas. **d** Schematic of the workflow.

impact of lesion sizes on the model performance, two subsets of the main test set were established. Each comprised all 861 negative LN images. One contained the 58 positive LN images demonstrating only micrometastasis (≥0.2 mm, <2 mm), and the other contained the 28 positive LN images demonstrating only ITCs (<0.2 mm). The model achieved AUCs of 0.9940 (0.9892–0.9988) and 0.9228 (0.8643–0.9814) on the micrometastasis and ITC test subsets, respectively. The results indicated that the ITC identification accuracy of the model remained to be improved (Fig. 2a).

To prepare the model for practical use, a threshold of prediction scores was set such that the model could generate a concrete prediction regarding whether an LN was positive or negative. In the general context, the threshold was set as 0.4 to balance positive and negative predictions, which acquired a

relatively high Matthews correlation coefficient (MCC; a reliable confusion matrix metric[27]) score on the validation set compared to other thresholds. Under a threshold of 0.4, the model achieved a sensitivity of 0.8915 (0.8503–0.9246), a specificity of 0.9861 (0.9758–0.9928), and an MCC of 0.8986 (0.8686–0.9269) on the main test set. These results are comparable to those of patch-based methods[14,15] (MCCs: 0.8937 and 0.9334). Table 1 and Supplementary Table 1 presents the performance of our model, the pathologists, and previous models[14,15] under the main test set as well as the micrometastasis and ITC test subsets. On the other hand, in the clinical context, where AI is used to screen suspicious LNs, a more sensitive threshold of 0.15 was employed.

**Comparisons with other weakly supervised methods.** The results indicated that the proposed ESCNN was accurate in

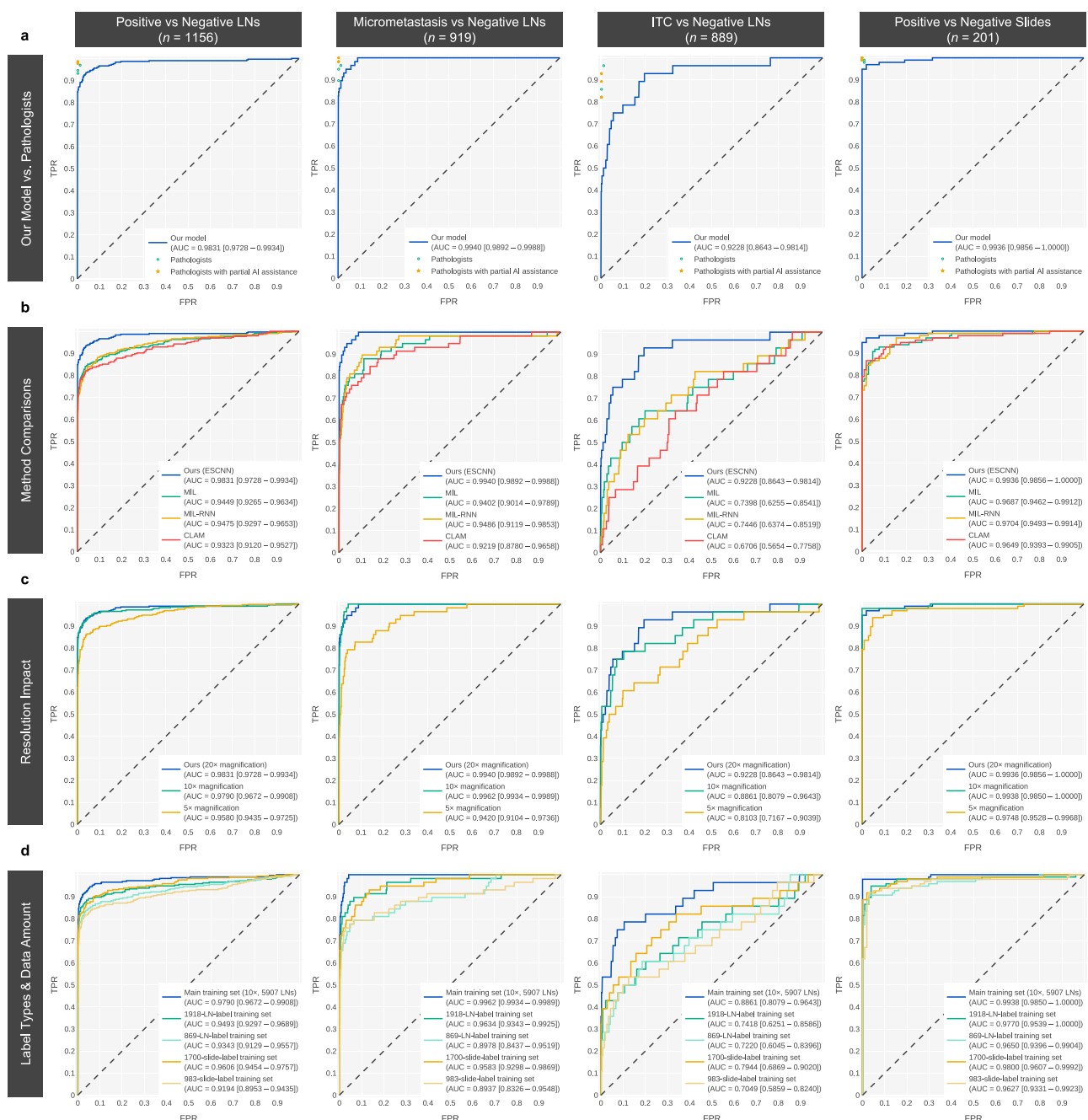

**Fig. 2 Receiver operating characteristic (ROC) curves of the metastasis identification ability of the models under the main test set of 1156 LN images.** The rows present a comparison of (**a**) the performance of our model with those of three pathologists before and after receiving AI assistance on the 48 equivocal cases; (**b**) weakly supervised methods; (**c**) performance obtained under various magnification levels (20×: 0.46 μm/pixel, 10×: 0.92 μm/pixel, 5×: 1.84 μm/pixel); and (**d**) performance obtained under different label types and amounts of training data. We evaluated each method with the main test set and its subsets to retrieve the ROC curves. The first column presents the ROC curves differentiating between the 1156 LNs in the main test set. Among the 1156 LNs, those marked as micrometastases and ITCs, as well as all the negative LNs, were sampled to evaluate the performance of the model in identifying micrometastases and ITCs, as presented in the second and third columns. The fourth column displays the slide-level performance.

identifying gastric LN metastasis under weak supervision. We next focused on investigating the performance of other alternatives. Most weakly supervised methods adopt a two-stage architecture: patch feature extraction and interpatch aggregation. MIL typically involves the use of a first-stage CNN for making patch predictions, followed by a second-stage selection of the most suspicious patch with the highest prediction score (i.e., through max pooling) to represent the entire slide. In an MIL–recurrent neural network (MIL-RNN) architecture[17],

instead of max pooling, an RNN is employed to aggregate the patch embeddings of top-scoring patches. Under the clustering-constrained attention MIL (CLAM)[18] approach, a pretrained CNN is leveraged to extract patch embeddings in the first stage. The second stage involves a clustering-constrained attention module. Under the main training set, MIL, MIL-RNN, and CLAM yielded AUCs of 0.9449 (0.9265–0.9634), 0.9475 (0.9297–0.9653), and 0.9323 (0.9120–0.9527) for LN image classification, respectively, and AUCs of 0.9687 (0.9462–0.9912),

**Table 1 Overview of the performance of our model, the pathologists, and previous models.**

| Model / Pathologist | TP | FP | FN | TN | Sensitivity | Specificity | MCC |
|---|---|---|---|---|---|---|---|
| Our model | 263 | 12 | 32 | 849 | 0.8915(0.8503–0.9246) | 0.9861(0.9758–0.9928) | 0.8986(0.8686–0.9269) |
| Pathologist S.-C.H. | 289 | 2 | 6 | 859 | 0.9797(0.9563–0.9925) | 0.9977(0.9916–0.9997) | 0.9818(0.9681–0.9932) |
| Pathologist 1 | 279 | 2 | 16 | 859 | 0.9458(0.9134–0.9687) | 0.9977(0.9916–0.9997) | 0.9589(0.9392–0.9767) |
| Pathologist 2 | 286 | 11 | 9 | 850 | 0.9695(0.9429–0.9860) | 0.9872(0.9773–0.9936) | 0.9546(0.9340–0.9731) |
| Pathologist 3 | 275 | 2 | 20 | 859 | 0.9322(0.8972–0.9581) | 0.9977(0.9916–0.9997) | 0.9497(0.9284–0.9690) |
| Pathologist 1with partial AI assistance | 289 | 2 | 6 | 859 | 0.9797(0.9563–0.9925) | 0.9977(0.9916–0.9997) | 0.9818(0.9682–0.9932) |
| Pathologist 2with partial AI assistance | 291 | 2 | 4 | 859 | 0.9864(0.9656–0.9963) | 0.9977(0.9916–0.9997) | 0.9863(0.9742–0.9956) |
| Pathologist 3with partial AI assistance | 288 | 2 | 7 | 859 | 0.9763(0.9517–0.9904) | 0.9977(0.9916–0.9997) | 0.9795(0.9648–0.9912) |
| Hu et al. | 159 | 11 | 21 | 1025 | 0.8833(0.8272–0.9263) | 0.9894(0.9811–0.9947) | 0.8937(0.8566–0.9283) |
| Wang et al. | 5217 | 391 | 82 | 9544 | 0.9845 (0.9808–0.9877) | 0.9606 (0.9566–0.9644) | 0.9334(0.9275–0.9391) |

The confusion matrices were calculated for our model (at a threshold of 0.4) and the pathologists, including the number of true-positive (TP), false-positive (FP), false-negative (FN), and true-negative (TN) LN images under the main test set ($n = 1156$). Three pathologists (J.L., H.-C.C., and T.-Y.H.) relabeled the 38 equivocal LN images with AI assistance (denoted as partial AI assistance). The data on model performance reported in the bottom two rows of the table were directly retrieved from the publications in question. Considering the between-study discrepancies in test slide distributions, the results may contain bias. MCC is an abbreviation for Matthews correlation coefficient. Supplementary Table 1 provides extended information, including additional metrics and model performance results on the micrometastasis and ITC test subsets.

0.9704 (0.9493–0.9914), and 0.9649 (0.9393–0.9905) for WSI classification, respectively. As shown in Fig. 2b, the proposed ESCNN model (AUC for LN image classification: 0.9831) empirically outperformed these alternatives ($P < .001$).

**Impact of image resolutions, data set size, and label types on ESCNN performance**. Aside from two-stage weak supervision, end-to-end training methods such as streaming CNN[24] and the whole-slide training method[22] also demonstrate favorable classification performance on tasks performed under low magnification. However, image resolution under these methods is constrained by the prohibitively low throughput and high memory consumption. Because all these end-to-end methods are logically equivalent, we used the ESCNN to evaluate their performance when applied to downsampled WSIs ($5\times$ and $10\times$ magnification). The AUCs of ESCNN models trained using LN images magnified $5\times$ (resolution: 1.84 μm/pixel) and using LN images magnified $10\times$ (0.92 μm/pixel) were 0.9580 and 0.9790, respectively, lower than the AUC of 0.9831 obtained using LN images magnified $20\times$ ($P = .001$ and $.35$). Further analysis revealed that the ability to identify micrometastases became saturated after application to micrometastasis subset images magnified $10\times$ (AUCs corresponding to LN images magnified $5\times$, $10\times$, and $20\times$: 0.9748 vs. 0.9938 [$P < .001$] vs. 0.9936 [$P = .35$]). By contrast, the ability to identify ITCs from the ITC subset improved continually with increasing image resolution (0.8103 vs. 0.8861 [$P = .044$] vs. 0.9228 [$P = .13$]). The benefits conferred by $10\times$ magnification were significant. However, the benefits of $20\times$ magnification required the verification of more ITC samples (Fig. 2c).

We also evaluated the impacts of data set sizes and the label types (slide level or LN level) on identification performance. Under training with LN-level labels and images magnified $10\times$, the identification performance was enhanced after a larger data set was input (AUC of LN image classification: 0.9343, 0.9493 [$P = .032$], and 0.9790 [$P < .001$] for the truncated 869-LN-image, truncated 1918-LN-image, and full 5907-LN-image data sets). The results suggest that the input of more training data improved model performance. Under training with only slide-level labels, the AUCs of LN image classification obtained using 983 and 1700 WSIs were 0.9194 and 0.9606, respectively. Notably, regardless of the label type, the model performance corresponded relatively well to the number of labels. The two models trained using 869 LN images and 983 WSIs achieved comparable results (LN-level AUC = 0.9343 vs. 0.9194, $P = .13$), as did the models trained using 1918 LN images and 1700 WSIs (LN-level AUC = 0.9493 vs. 0.9606, $P = .11$). In short, when the total number of slides is limited, LN-level labels are recommended for enhancing model performance (Fig. 2d).

**Throughput and memory consumption**. Despite the logical equivalence of these end-to-end training methods, the vast computational and memory overhead involved precludes the handling of high-resolution tasks. As presented in Fig. 3, we examined the throughputs and memory consumptions of these approaches under various input image resolutions ($4688 \times 4688$ [$1.25\times$], $9375 \times 9375$ [$2.5\times$], $18,750 \times 18,750$ [$5\times$], $37,500 \times 37,500$ [$10\times$], and $75,000 \times 75,000$ [$20\times$]) by using 100 randomly sampled LN images from the main training set. Among these image resolutions, an original ResNet50[26] model can undergo direct end-to-end training only on $1.25\times$ images (memory consumption: 19.1 GB) due to limited GPU memory capacity (NVIDIA Tesla V100 with 32 GB of random-access memory [RAM]). The whole-slide training method[22] leverages CUDA Unified Memory to enable the excessive amount of intermediate data stored in GPU memory to be offloaded to host memory through data swapping. Although host memory is $10\times$–$100\times$ larger than GPU memory on a typical GPU server, this method can train a $5\times$ model (memory consumption: 618.9 GB) on a server with 768 GB of system memory at best. Moreover, the throughput was considerably hindered (0.153 images per minute for training on $5\times$ images) by the overhead incurred by GPU–host memory data transfer. By contrast, the streaming CNN and ESCNN methods reduced the amount of intermediate data, such that the memory consumption for model training remained between 8 and 9 GB regardless of the image resolution. This ensured that all the intermediate data could fit into the GPU memory, thus obviating the need for Unified Memory. Without the data swapping overhead, the training throughputs of streaming CNN and ESCNN on a $5\times$ model training were 1.49 and 3.48 images per minute, which were $9.74\times$ and $22.7\times$ faster than the whole-slide training method, respectively. When trained on $20\times$ images, the ESCNN approach obtained a training throughput of 0.912 images per minute, which was $9.83\times$ faster than the 0.0928 images per minute achieved under the streaming CNN method. This improvement is attributable to the patch-based image augmentation ($2.31\times$ speedup) and skipping mechanism ($4.26\times$ speedup) under the ESCNN approach.

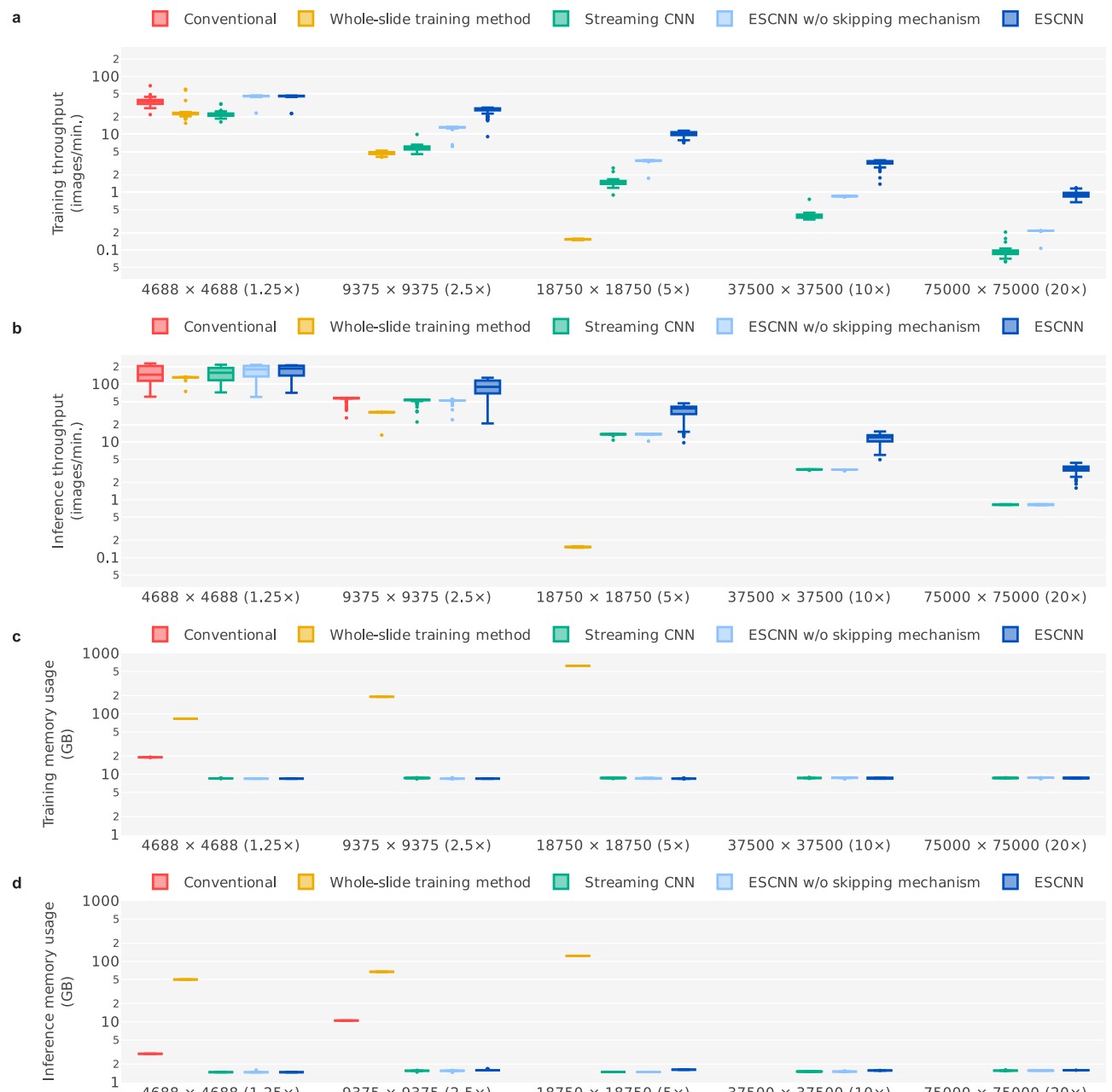

**Fig. 3 Throughput and memory consumption of end-to-end training methods under various magnification levels of LN images.** Each panel represents the (**a**) training throughput, (**b**) inference throughput, (**c**) training memory consumption (referring to Unified Memory for the whole-slide training method and GPU memory for the others), and (**d**) inference memory consumption. For each setting, we recorded the training/inference time and memory consumption when processing each LN image ($n = 100$ images in total, sampled from the main training set). Each box-and-whisker plot comprises the center (median), the bounds of boxes (Q1 and Q3), the bounds of whiskers (the minimum and maximum within the range, obtained by adding the median to ±1.5 times the Q3–Q1 distance), and the outliers of the underlying 100 samples. The absence of certain boxes indicates that those settings could not be run due to memory shortages.

**Lesion highlights and qualitative analysis.** The model highlighted metastatic tumor areas for rapid verification through class activation mapping (CAM)[28]. In quantitative analysis, the saliency maps generated by the algorithm achieved an Intersection over Union (IoU) of 0.5934 (at a threshold of 0.5) and a pixel-level AUC of 0.8495 in five detailed-annotated WSIs sampled from the main test set, demonstrating high correspondence between the predicted and actual lesion areas.

As displayed in Fig. 4, the CAM results of our model exhibited a higher coverage of macrometastases and micrometastases, and the ability to localize ITCs, compared to the other methods.

Furthermore, CAM was employed to investigate the sources of false predictions of our model (Fig. 5). Specifically, 24 false-positive slides were reviewed. Slides with artifacts (including cautery, crushing, and floater artifacts; 5, 21%) and histiocytic aggregates (3, 13%) may have misled the model. No common patterns were found in the remaining 16 false-positive slides. Within the reviewed 13 false-negative slides, the metastatic foci were mostly ITCs (11, 85%) and micrometastases (2, 15%). As for morphology, most cases were classified as diffuse or mixed-type adenocarcinoma (10, 77%), characterized by low numbers of dispersed ITCs that may have resembled sinus histiocytes in

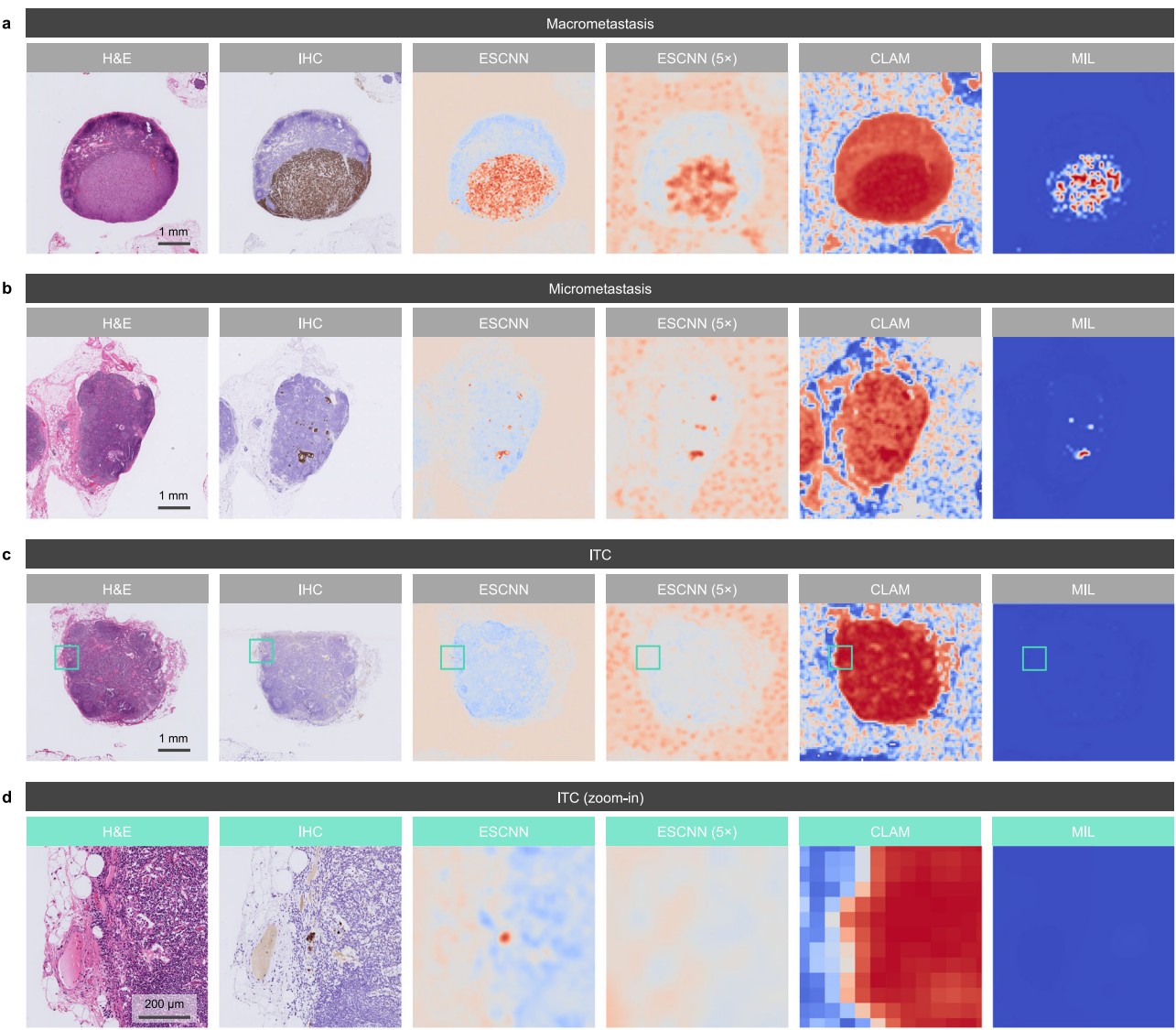

**Fig. 4 Qualitative results of lesion highlights on macrometastasis, micrometastasis, and ITC cases.** Each panel displays an example of an H&E-stained LN image, the reference standard of the metastatic area under IHC staining (cytokeratin AE1/AE3), the heat map for lesion localization generated by our model through CAM, the heat map generated by a 5× low-resolution ESCNN model, the attention map of a CLAM model, and the prediction map of a MIL model. Identified metastatic tumor cells are highlighted in brown and red in the IHC stains and heat maps, respectively. Examples of (**a**) macrometastasis, (**b**) micrometastasis, and (**c, d**) ITCs identified from the main test set demonstrated the high correspondence of the model-predicted area with the IHC-predicted area, where (**d**) displays the high-power field of the green boxes in (**c**). Aside from the displayed examples, the localization performance of our model remained at the same level for the 263 LN images from the main test set that were correctly classified as metastases.

appearance. The remaining cases (3, 23%) corresponded to intestinal-type adenocarcinoma, in which ITCs with clear cytoplasm were observed.

**Comparisons with pathologists and a pilot study of AI assistance.** The main test set was reviewed by four pathologists. The classification of each LN image by each pathologist was examined. The MCCs of the four pathologists (0.9497–0.9818) exceeded that of the model (0.8986). Notably, regarding the 48 LN images for which consensus among the four pathologists was not reached (sensitivity: 39.4–81.8%; specificity: 26.7–86.7%), the model exhibited a relatively high sensitivity of 69.7% and specificity of 86.7%. In other words, AI assistance was helpful under this equivocal situation. To confirm this premise, three pathologists (J.L., H.-C.C., and T.-Y.H.) were asked to double-review the 48 equivocal LN images by using the AI-assisted LN assessment workflow. Overall, 42.4% of the previous labels were changed, and

the performance of all three pathologists improved (MCCs without assistance: 0.9497–0.9589, MCCs with assistance: 0.9795–0.9863) to the level of that of the expert pathologist (S.-C.H.; 0.9818). Therefore, we proceeded to conduct a formal study for validating the clinical impact of the AI-assisting workflow in terms of review time, accuracy, and count consistency.

**Clinical impact of the AI-assisted LN assessment workflow.** As mentioned, the assessment workflow included an LN detector. Trained using the main training set of 5907 LN images, the DeepLabv3 + -based[29] LN detector achieved an IoU of 0.8473 and a pixel-wise accuracy of 92.83%. Six pathologists (J.L., T.-Y.H., H.-C.C., K.-H.C., R.-C.W., and Y.-J.L.) were recruited to review 80 slides with and without AI assistance, with a 2-to-3-week washout interval. The slides, sampled from the archive of Linkou CGMH in 2020, comprised 19 negative slides, 24 slides

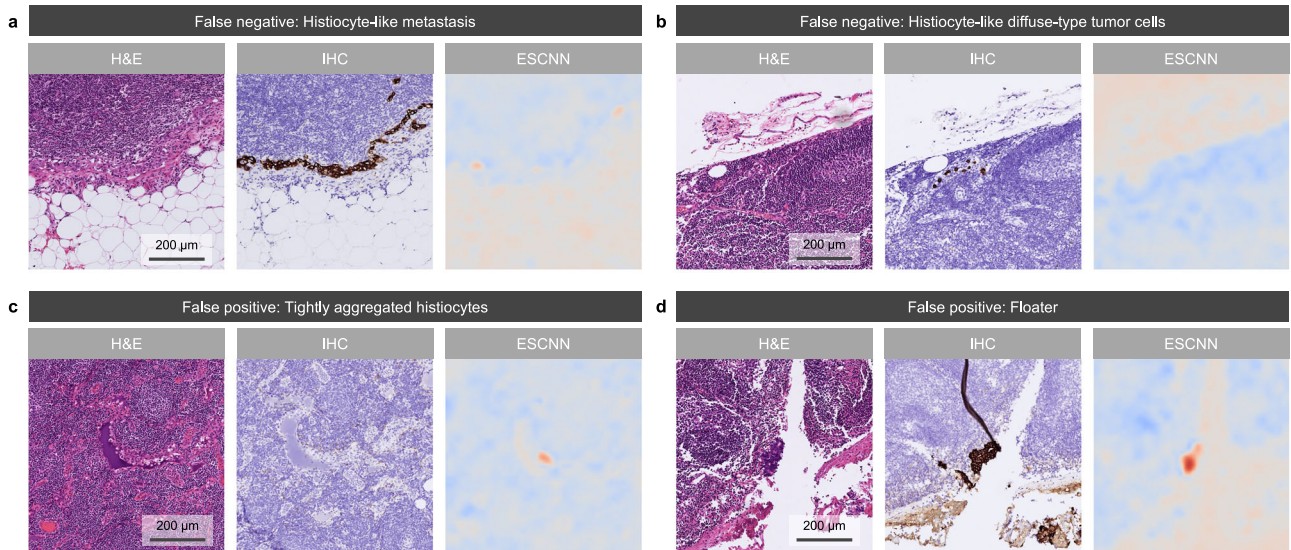

**Fig. 5 Qualitative results of lesion highlights on false-negative and false-positive cases.** Each panel displays an example of an H&E-stained LN image (left), the reference standard of the metastatic area under IHC staining (cytokeratin AE1/AE3; middle), and the heat map for lesion localization generated by our model through CAM. **a** Example of false-negative cases indicating histiocyte-like metastasis, which tended to mislead our model. The main test set contained 10 similar samples. **b** Histiocyte-like diffuse-type ITCs in sinusoids were challenging for both the model and the pathologists; their accurate detection may require IHC slides. **c** Slide showing histiocytic aggregates in sinusoids and unusual blue proteinaceous fluid that caused our model to issue a false alarm, which led five of the six pathologists to incorrectly interpret the slide as positive. The incorrect highlight of histiocytic aggregates appeared in three samples. **d** Floater misidentified by our model and five of the six pathologists under AI assistance as metastatic adenocarcinoma. Five slides with artifacts (including cautery, crushing, and floater artifacts) were misidentified.

with macrometastasis (≥2 mm), 24 slides with micrometastasis (<2 mm, ≥0.2 mm), and 13 slides with ITCs (<0.2 mm).

As indicated in Fig. 6a, the workflow significantly shortened the review time of the pathologists (per-slide median: 161.2 to 110.5 s, −31.5%, $P < .001$). The review time also significantly decreased for negative slides (178.9 to 127.7 s, −28.6%, $P < .001$), slides exhibiting macrometastasis (166.7 to 112.2 s, −32.7%, $P < .001$), slides exhibiting micrometastasis (142.2 to 99.4 s, −30.1%, $P < .001$), and slides exhibiting ITCs (139.7 to 103.4 s, −26.0%, $P = .005$). AI assistance accelerated the classification of most cases. In a few negative cases, however, the review time increased slightly. As presented in Fig. 6b, mixed-effect modeling revealed that AI-attributable false alarms affected the time taken to review negative LN images (median time for slides with and without false alarms: 146.3 vs. 119.1 s, respectively; $P = .047$). In short, the AI-predicted false positives prompted the pathologists to scrutinize the slides of interest more thoroughly, thus increasing the review time. However, the time taken remained shorter than that under no AI assistance.

Regarding the accuracy of reported positive LN slide (with a classification of positive meaning that at least one positive LN was detected), under AI assistance, the slide-level sensitivity increased significantly from 81.94% (79.25–92.18%) to 95.83% (91.15–98.46%, $P < .001$) for the slides exhibiting micrometastases. For the slides exhibiting ITCs, the slide-level sensitivity improved significantly from 67.95% (56.42–78.07%) to 96.15% (89.17–99.20%, $P < .001$). The sensitivity corresponding to the slides exhibiting macrometastasis remained at the same high level without (99.31% [96.19–99.98%]) and with (100.0% [97.47–100.0%], $P > .99$) AI assistance (Fig. 6c). As displayed in Fig. 6d, in some cases, false alarms in negative slides resulted in the pathologists reporting false-positive results, causing the specificity to drop from 93.86% (87.76–97.50%) to 84.21% (76.20–90.37%, $P = .019$). All but one false alarm (16/17) was concentrated in 3 of the 19 negative slides. They misled five to six of the pathologists. The class activation maps of these slides

highlighted regions containing tightly aggregated histiocytes with unusual blue proteinaceous fluid, increased numbers of high endothelial venules, and unintentionally introduced floater artifacts, respectively (Fig. 5).

The counts of positive LNs differed among the pathologists. This is ascribable to inconsistent diagnoses of LN metastasis and to variations in subjective distinctions of LN and non-LN tissue. Under AI assistance, the consistency of positive reports, as indicated by the coefficient of variation (CV; a lower value is desirable), increased significantly (median: 0.3499 to 0, $P < .001$) in all the positive categories, namely macrometastasis (0.1775 to 0, $P < .001$), micrometastasis (0.3651 to 0, $P < .001$), and ITCs (0.6388 to 0.1113, $P = .014$; Fig. 6e). The consistency of negative reports increased as well, but not as markedly (Fig. 6f).

**Cross-site evaluation.** After the assessment of both the performance of the ESCNN model and the clinical experiment using slides from Linkou CGMH, we validated the robustness of the workflow. Specifically, we applied the 20× ESCNN model to the 327 slides collected from Kaohsiung CGMH between 2019 and 2021 with the 2088 LN images annotated by S.-C.H., J.L., H.-C.C., T.-Y.H., and K.-H.C. Regarding the cross-site performance of metastasis identification, AUCs of 0.9868 (0.9784–0.9952) and 0.9829 (0.9652–1.0) were achieved for the classification of LN images and WSIs, respectively. These AUCs were not significantly different from those of the main test set (0.9831 [$P = .59$] and AUC = 0.9936 [$P = .29$], respectively). The IoU of 0.9044 of the LN detector (vs. 0.8522 on the main test set) also indicated high model generalizability.

## Discussion
Herein, we applied the ESCNN approach to the direct end-to-end training of models on high-resolution images (i.e., images at 20× magnification) with LN-level and slide-level labels. ESCNN enhanced the performance of the weakly supervised model to the

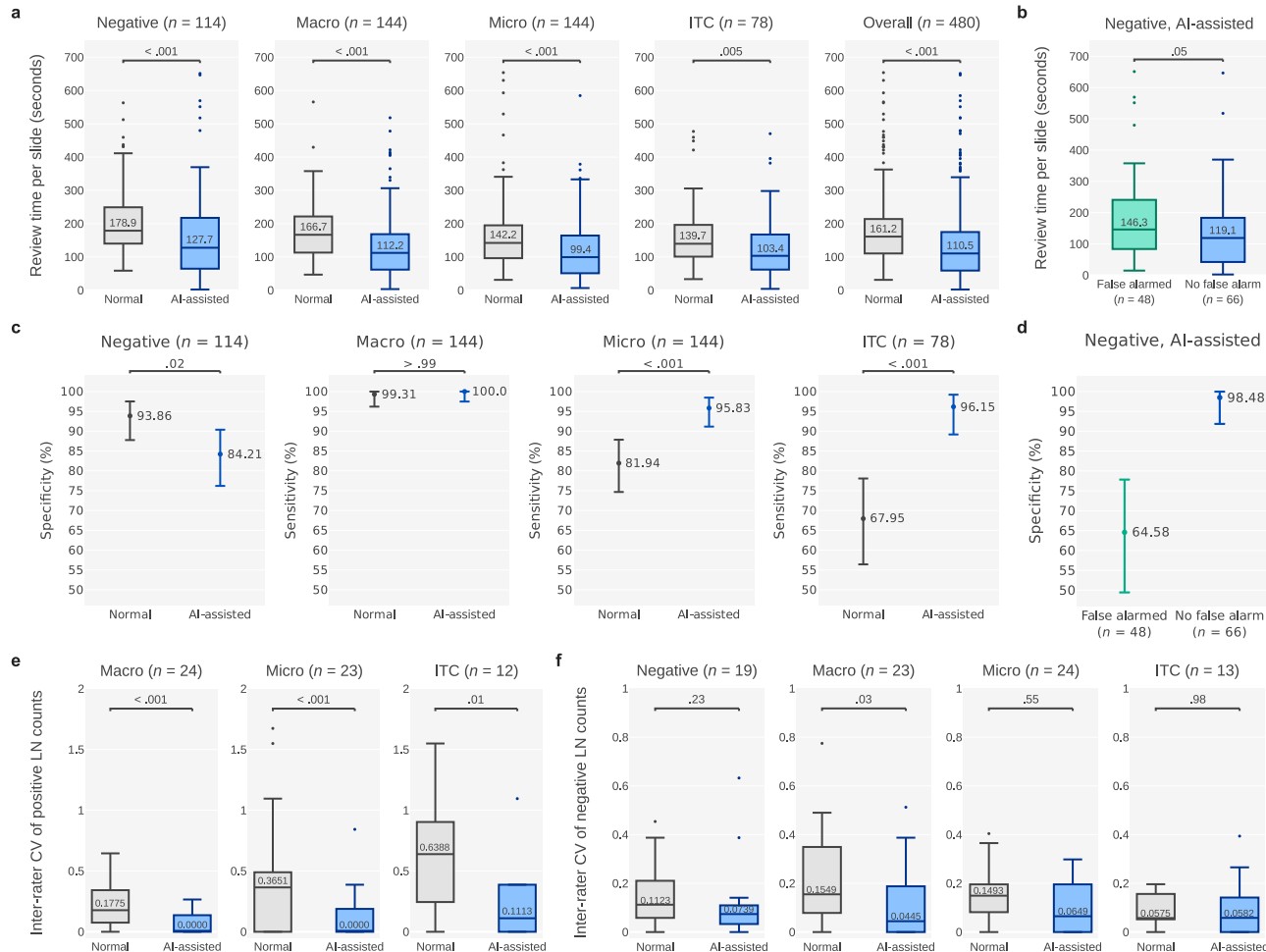

**Fig. 6 Plots showing the review time, accuracy, and LN count consistency in the clinical experiments. a** Per-slide review time with and without AI assistance. Macro and Micro are the abbreviations of macrometastasis and micrometastasis respectively. **b** Impact of AI-attributable false alarms on review time, assessed by comparing the time taken to review negative slides for which false alarms were issued with the time taken to review negative slides for which false alarms were not issued. **c** Accuracies (i.e., specificities for negative slides and sensitivities for positive slides) achieved with and without AI assistance. **d** Impact of AI-attributable false alarms on specificities. **e** CVs per slide, calculated using the positive LN classifications of the six pathologists to quantify the interrater reliability of positive LN counts. **f** CVs per slide of negative LN classifications. The box-and-whisker plots in (**a**), (**b**), (**e**, **f**) comprises the center (median), the bounds of boxes (Q1 and Q3), the bounds of whiskers (the minimum and maximum within the range, obtained by adding the median to ±1.5 times the Q3–Q1 distance), and the outliers. The numbers within the boxes are the medians. The centers and error bars in (**c**, **d**) represent the sensitivities (or specificities) and the 95% confidence intervals, respectively.

level of patch-based models with significantly less annotation effort[14,15]. Moreover, the implementation of the AI-assisted LN assessment workflow in routine pathological practice is expected to facilitate the acquisition of LN-level labels. Given a new slide image, reviewed LN contours can directly be fed as LN annotations for model finetuning. A vast amount of new training data can be efficiently obtained with this mechanism (e.g., 1300 slides per year in Linkou CGMH). We expect that the performance of the metastasis identification model can be further enhanced using labeled images accumulated in a routine clinical workflow.

The clinical experiment revealed that the AI-assisted LN assessment workflow confers benefits to clinical practice with respect to the review time, sensitivity, and consistency. These results accord with those of studies examining the impacts of AI assistance in pathological analysis. Steiner et al.[12] developed a metastatic breast cancer identification model that increased the sensitivity with which micrometastasis was identified from 83% to 91% and reduced the review time for both micrometastasis (−47.4%) and negative cases (−19.0%). Kiani et al.[30] reported that AI assistance in the classification of liver cancer significantly

improved the accuracy ($P = 0.045$, odds ratio [OR] = 1.499). The model developed by Zhou et al.[31] significantly shortened the time taken to review *Helicobacter pylori*–positive cases ($P = 0.003$) and significantly increased the sensitivity with which they were identified (OR = 13.37). Taken together, the evidence indicates that AI-assisted workflows help reduce the review time and enhance sensitivity. We estimate that the present workflow can save approximately 1000 min of pathologists' time each year. This was calculated according to the total number of slides (1294) collected by Linkou CGMH in 2019 and according to the average reduction in the per-slide review time (50.7 s). However, we observed that AI-attributable false alarms misled the pathologists on negative results, reducing the specificity from 94% to 84%. This phenomenon was also observed in a study by Zhou et al.[31], in which AI-attributable false alarms reduced the specificity corresponding to the classification of cases negative for *H. pylori* (OR = 0.435). Notably, regarding positive LN images, the significantly improved sensitivity and interrater stability suggest that pathologists can reach a greater consensus in clinical practice with AI assistance. Some investigators argued that

micrometastasis in gastric cancer should be reflected in the N category[6]. The February 2021 update to the National Comprehensive Cancer Network guidelines recommends adjuvant chemotherapy as treatment for patients with pT1N0 or pT2N0 gastric cancer with micrometastases or ITCs[10]. However, given the conflicting evidence on the subject, the prognostic impact of detecting micrometastases and ITCs remains debatable[8,9]. This controversy may be due to the difficulty in collecting a sufficiently large number of ITCs. Moreover, in view of the challenges involved in detecting micrometastases and ITCs in clinical scenarios based on slides stained with hematoxylin and eosin (H&E), the possibility of between-patient treatment discrepancies should be considered. The ignorance of such differences may constitute a confounding factor that reduces the validity of studies. Employment of the AI-assisted workflow is expected to resolve these problems. Its high detection sensitivity enables the selection of more cases of interest, allowing all patients to receive suitable treatment. In turn, the real significance of detecting micrometastases and ITCs to the prognosis can be revealed. On the other hand, the drop in specificity in the present study requires further consideration. Errors were concentrated in three slides, with some AI-highlighted areas that resembled metastatic tumor cells being identified as ITCs by five to six pathologists. Besides, the high number of micrometastasis and ITC cases in the test set (30% and 16%, respectively) could contribute to the decline in specificity. Such high proportions of micrometastasis and ITC cases were included to ensure sufficient numbers of samples per category. These characteristics of the data set resulted in the tendency of the pathologists to lean toward AI predictions rather than relying on their interpretation. Specifically, they favored positive assessments although the slides were negative. False alarms would probably have a weaker impact in clinical practice, where the categories in data sets are naturally distributed. In the diagnosis of challenging cases, the pathologists are provided tools such as IHC staining[32,33] and primary tumor reference images. Given the distinct histological appearances of primary tumors and metastatic regions corresponding to false alarms, the three slides of interest should have been classified correctly. We suggest employing cytokeratin IHC to verify cases involving shifts in AJCC staging, especially shifts from N0 to N1 status. In the near future, we will conduct follow-up research on the clinical application of the workflow to determine whether this concern of specificity drop persists. The examination of prognostic impacts derived from shifts in the N category is recommended for future studies. Moreover, we expect that the false alarm problem can be resolved by improving the metastasis identification model. To reach this target, further scaling up of the training set is highly recommended because model performance has not reached saturation. The present workflow is effective in reducing efforts devoted to the collection of node-level labels. The final obstacle to model optimization is the presence of noisy labels in training data[34,35]. Specifically, the model can be misled by pathologists' inability to locate all ITCs. This inability is attributable to the high percentage of diffuse and mixed-type cancer in gastric cancer[3]. The forward correction of cross-entropy loss[35] that we employed mitigated this problem, but not significantly. This concern may be resolved completely by introducing a noisy label correction mechanism[36] that routinely identifies possible labeling errors in data sets for IHC verification or further expert review.

This study is associated with some possible future works. First, the ESCNN method divides the training procedure into subtasks through spatial partition and solves them independently to save memory. Such a design conflicts with operations requiring a global context, including batch normalization[37], Squeeze-and-Excitation[38], and Vision Transformer[39]. One possible solution is to combine the ESCNN approach with halo exchange[40], where

subtasks are executed concurrently using a sufficiently large number of computing devices. Inter-patch information can be exchanged through inter-device communication. Second, the ESCNN method is currently employed in classification, but it can be extended to conduct semantic segmentation or detection tasks in the future. Strongly supervised segmentation and detection model architectures are not the targets of interest for the ESCNN approach because these models require detailed annotations for training, with which the image resolution issue can be resolved simply through patching. Applying weakly supervised semantic segmentation (WSSS) and detection (WSOD) to medical images is relevant to the ESCNN method. Numerous WSSS and WSOD architectures adopt a two-stage workflow comprising a first-stage classifier to produce pseudo annotations and a second-stage strongly supervised segmentation or detection model to learn from those pseudo annotations[41,42]. In such architectures, the ESCNN approach can be employed as a classifier for producing pseudo annotations through CAM to realize whole-slide WSSS and WSOD. Third, ESCNN leverages the gradient checkpointing[43] technique to reduce memory consumption. This increases the overhead for recomputing feature maps. Even if we considered this as a cost-effective trade-off in view of the vast throughput leap conferred by memory saving (22.1× speedup against the whole-slide training method[22]), reducing this overhead remains a necessary endeavor.

Since the success of the CAMELYON16 challenge of breast cancer LN metastasis identification in 2016[11], no commercial deep learning software has been developed for identifying LN metastasis. This reflects not only the time demand for the generalization of AI-enabled software but also the difficulty in establishing a reliable clinical-grade deep learning algorithm for identifying LN metastasis, especially high morphological variations within the same anatomic site for different cancers. The embedded metastasis identification model was trained through the proposed ESCNN and outperformed all other models through its use of end-to-end, high-resolution deep learning. The results of the clinical experiment demonstrated the improvement of the pathologists' accuracy in identifying micrometastases and ITCs and the shortening of the review time. Further large-scale or multicenter studies are warranted to test the practicability of the workflow. The improvement to the N category evaluation is expected to confer prognostic benefits and indicate that AI promotes enhancements in healthcare quality.

## Methods

**Samples and slide images**. We retrospectively retrieved slides featuring surgical resections of LNs from gastric carcinoma dissections between 2018 and 2020 from the archive of the Department of Anatomic Pathology at Linkou CGMH and Kaohsiung CGMH. In total, we examined 422 studies, 2422 H&E slides, 125 immunohistochemistry (IHC) slides, and approximately 20,000 LN images. The WSIs were obtained and digitalized at 40× magnification (0.23 μm/pixel) on a NanoZoomer S360 digital slide scanner (Hamamatsu Photonics, Hamamatsu, Japan). The study protocol was approved by the Institutional Review Board (IRB) of CGMH (IRB No. 202000038B0). Written informed consents were waived by the IRB in the study due to the usage of deidentified digital slides.

**Data preparation for model training and evaluation**. The slides were categorized according to their purpose. The main test set, which was used to evaluate and compare the model's performance with that of the pathologists, comprised 30 studies, including 201 slides obtained through total population sampling from the archive of Linkou CGMH in 2019. Total population sampling was employed to prevent sampling bias. To obtain the reference standard, each slide in the test set was repeatedly annotated by four certified pathologists (S.-C.H., J.L., H.-C.C., and T.-Y.H.) through the aetherSlide digital pathology system (aetherAI, Taipei, Taiwan). The LNs on each slide were contoured and labeled as either negative or positive for metastatic carcinoma. Micrometastases or ITCs were marked non-exhaustively for analysis. In total, 1156 LN contours were labeled with four individual classifications, after excluding 149 contours on which agreement could not be reached regarding whether each contour encompassed LN or non-LN tissue. Based on H&E staining results, the consensus was established for 1108 of the 1156

LNs. The 48 slides for which consensus was not reached were further stained for IHC testing with cytokeratin (clone AE1/AE3, 1:200, Genemed, Torrance, CA, USA), Ep-CAM (clone BerEP4, 1:100, Cell Marque, Rocklin, CA, USA), and calretinin (clone Poly, 1:100, Genemed, Torrance, CA, USA), to differentiate metastatic carcinoma from intranodal mesothelial cells. The immunohistochemical procedures were conducted in an automated immunostaining machine (BOND-MAX, Leica Microsystems) with optimal negative and positive controls according to the manufacturers' protocols. Specifically, a slide of control tissue is run in every staining batch. The control tissue is derived from the human species and contains both positive and negative staining cells (epithelial cells for cytokeratin, adenocarcinoma for Ep-CAM, and mesothelial cells for calretinin) to serve as both the positive and negative controls. S.-C.H. made the final judgments after reviewing the IHC slides and the other pathologists' classifications.

The main training and validation sets consisted of 144 studies (983 WSIs, 5907 LN images) and 16 studies (110 WSIs, 655 LN images), respectively, collected from the archive of Linkou CGMH in 2019. The data sets excluded studies sampled for testing. The annotation process was similar to that for the test set—without interannotator validation. To increase the annotation efficiency by obviating the need for annotators to outline LNs, the LN detector generated editable LN contours for unannotated slides. An expanded training set incorporated 92 studies (718 WSIs) from the archive of Linkou CGMH in 2018. This set served as the training data of the model trained with slide-level labels to discuss the impact of label types. Each slide was marked according to whether LN metastasis was present. Tumor cells outside LNs, such as subserosal non-nodal extensions or free-floating cancer cells were ignored. The collection of this vast amount of data consolidates the robustness of the deep neural networks on which the present workflow is grounded.

**Overview of the gastric LN assessment workflow.** Automatically detecting positive and negative LNs is a classic instance segmentation problem for which sophisticated model architectures such as HTC[44] and DetectoRS[45] have been proposed in recent years. Those algorithms achieved favorable performance when applied to natural images, but their feasibility and effectiveness on WSIs are debatable. Directly training an instance segmentation model on WSIs is not feasible because of the high memory consumption attributable to the high resolutions of the images. Although patch-based methods resolve this problem in high-resolution image classification, they are not applicable to instance segmentation model training because instances in training images are required to remain whole without patching. Image downsampling is a satisfactory workaround for LN detection but sacrifices the model's ability to identify metastasis. To avoid this trade-off, we separated the proposed workflow into two modules, an LN detector and a gastric LN metastasis identification module, working at 1.25× and 20× magnification levels (7.36 and 0.46 μm/pixel), respectively. As the name suggests, the LN detector detects the boundaries of LNs. Large receptive fields of low magnification are preferred to capture comprehensive information about the LNs. The metastasis identification model operates under high magnification to identify subtle patterns in metastases, especially in ITCs.

**LN detector.** The LN detector is based on DeepLabv3 + [29], a semantic segmentation model that adopts atrous spatial pyramid pooling to substantially enlarge the receptive field such that the detector is capable of capturing the features of LNs of diverse sizes. The input of the detector is a WSI at 1.25× (7.36 μm/pixel) magnification and the output is a binary mask of equal size. The value of each pixel on the mask represents either the LN area (positive) or the background/area featuring non-LN tissue (negative). During the inference phase, predicted masks are converted into contours for LN enumeration. First, the positive area on the mask is expanded by morphological dilation using a 3 × 3 diamond-shaped kernel. Second, areas containing fewer than 4096 pixels are determined to be noise and removed. Finally, instances of LNs are segmented using Suzuki's method[46].

**ESCNN for gastric LN metastasis identification.** The gastric LN metastasis identification model classifies each LN instance into either positive (i.e., metastatic) or negative and highlights highly tumor-relevant areas to assist pathologists in rapid verification. In pursuit of precision, we developed an ESCNN for the direct end-to-end training of a classification model on high-resolution LN images and LN-level or slide-level labels, as illustrated in Fig. 7a.

The primary challenge of direct WSI training is that excessive memory usage cannot fit in the limited memory capacity of a graphics processing unit (GPU)[22]. Such high image resolution increases the memory requirement of each step in a typical training workflow, including image loading and image augmentation, as well as the backpropagation algorithm for updating a CNN. Although the elevated memory usage in image loading and image augmentation may cause no problem given sufficient host memory capacity, the limited GPU memory prevents the computation of a backpropagation algorithm on the GPU. A CNN consists of a stack of layers, most of which transform an input two-dimensional feature map (e.g., an RGB image) into another two-dimensional feature map. To update a CNN with a training pair comprising an image and its label, the backpropagation algorithm first conducts a forward pass, in which the image goes through layers to obtain the final prediction. Subsequently, the algorithm performs a backward pass,

in which the error between the prediction and the label propagates layers in reverse order. All the intermediate feature maps generated in the forward pass are retained until they are used in the backward pass, but their sizes increase in proportion to the image resolution and eventually exceed the memory capacity of the GPU. To reduce GPU memory consumption, the ESCNN follows the technique employed in streaming CNNs[24], involving the use of patching and gradient checkpointing. Both the forward and backward passes are divided into subtasks in the spatial dimensions. Each subtask handling a small region is sequentially executed to obtain a partial result, which is collected and then compiled with the others to obtain the full result. This divide-and-conquer technique saves memory space. Instead of the reuse of intermediate feature maps in the typical backpropagation algorithm, patches of intermediate feature maps generated in the forward pass are released and recomputed in the backward pass. This process is called gradient checkpointing[47]. By controlling the patch size such that each subtask is within the memory capacity of the GPU, this technique enables images of varying sizes to be processed on a GPU.

Aside from the memory capacity issue, the training throughput decreased sublinearly as the image size increased, hindering the application of gradient checkpointing to demanding tasks requiring larger images (e.g., metastasis detection). This performance bottleneck was observed during image augmentation, which entails the sequential application of processing steps to an image to increase data variety. The extremely high memory footprint involved in transforming a high-resolution image into another in each processing step increases paging overhead and memory loading (i.e., thrashing) considerably. To resolve this problem, the ESCNN introduces patching to image augmentation such that the locality of the memory access pattern can be increased to maximize the efficiency of the memory system. The implementation of patching is trivial for structure-preserving augmentations (e.g., color transformation) because the location of a patch is preserved after the augmentation. However, patching in morphological augmentations (e.g., rotation and scaling) faces a challenge due to the structural changes. Herein, patching in common morphological augmentations was facilitated by the patch-based affine transformation algorithm described in the Supplementary Information. Given that patching is conducted in both the image augmentation and successive backpropagation steps, a producer-consumer model, as illustrated in Fig. 7c, was constructed between the location where a patch buffer filled by an image processing thread is consumed by a backpropagation thread. Furthermore, the ESCNN implements a skipping mechanism to forgo unnecessary computations to achieve superlinear throughput scaling. All processing steps (image augmentation, forward pass, and backward pass) are skipped for contentless patches (pure white patches in our implementation). Patches with no gradient contribution to model parameters are dropped in the backward pass. Overall, this training pipeline resolves the thrashing issue by incorporating patching into data augmentation; skips unnecessary computations; and thus accelerates the overall throughput. Using the ESCNN training pipeline, we trained the metastasis identification model of the ResNet50 architecture[26] with two modifications. First, due to the inability of streaming CNN to tackle inter-patch computations, batch normalization layers were retained in the evaluation mode, performing linear transformations using two trainable parameters, namely γ and β. Second, the global average pooling layer was replaced by a global maximum pooling layer to handle the extremely small size of lesions presented on WSIs[22]. LN contours predicted by the LN detector were piped to this gastric LN metastasis identification module to yield a metastatic prediction—specifically, to label the contours as positive or negative. Furthermore, we used CAM[28] to generate highlights of the cancerous area by feeding the WSIs into the model. The feature map preceding the global pooling layer was extracted. Subsequently, a linear combination was performed with the weight in the final dense layer, and a sigmoid function was applied to output a [0, 1] score map. The score map was then scaled and transformed into a translucent mask for overlay on the original WSI.

**Model training.** We used the main training set, consisting of 983 WSIs and 5907 LN images, to train both the LN detector and the metastasis identification model on 32 NVIDIA Tesla V100 GPUs running in parallel. The weights in both the models were initialized as the pre-trained weights trained by the ImageNet data set[48]. Each training step dispatched one image per GPU. Images at 40× magnification were downscaled by 1/32 and 1/2 to 1.25× and 20× magnification to train the LN detector and the metastasis identification model, respectively. Random flip, random rotation (−180°–180°), stain matrix perturbation[49] (−10°–10°; stain matrix obtained by Vahadane et al.[50]), and random stain concentrations[49] (0.5×–1.5×) were applied to diversify the images before they were fed iteratively into the training pipeline. The LN detector was updated through stochastic gradient descent (initial learning rate: 0.01, momentum: 0.9, L2 decay: 0.0005) to minimize the per-pixel cross-entropy. After 40,000 training iterations with cosine annealing[43], the model parameters were saved. To update the metastasis identification model, we employed the AdamW optimizer[51] (initial learning rate: 1e−5, beta: [0.9, 0.999], weight decay: 0.01). The loss function, forward correction cross-entropy[35], mitigated the impact of incorrect labels with the transition matrix estimated from the labels of each pathologist and ground truths of the test set. Validation losses were evaluated after each training epoch. If no improvement was observed during 16 consecutive epochs, the learning rate was tuned to 1e−6, and the model continued training another 16-epoch stall. The optimal model

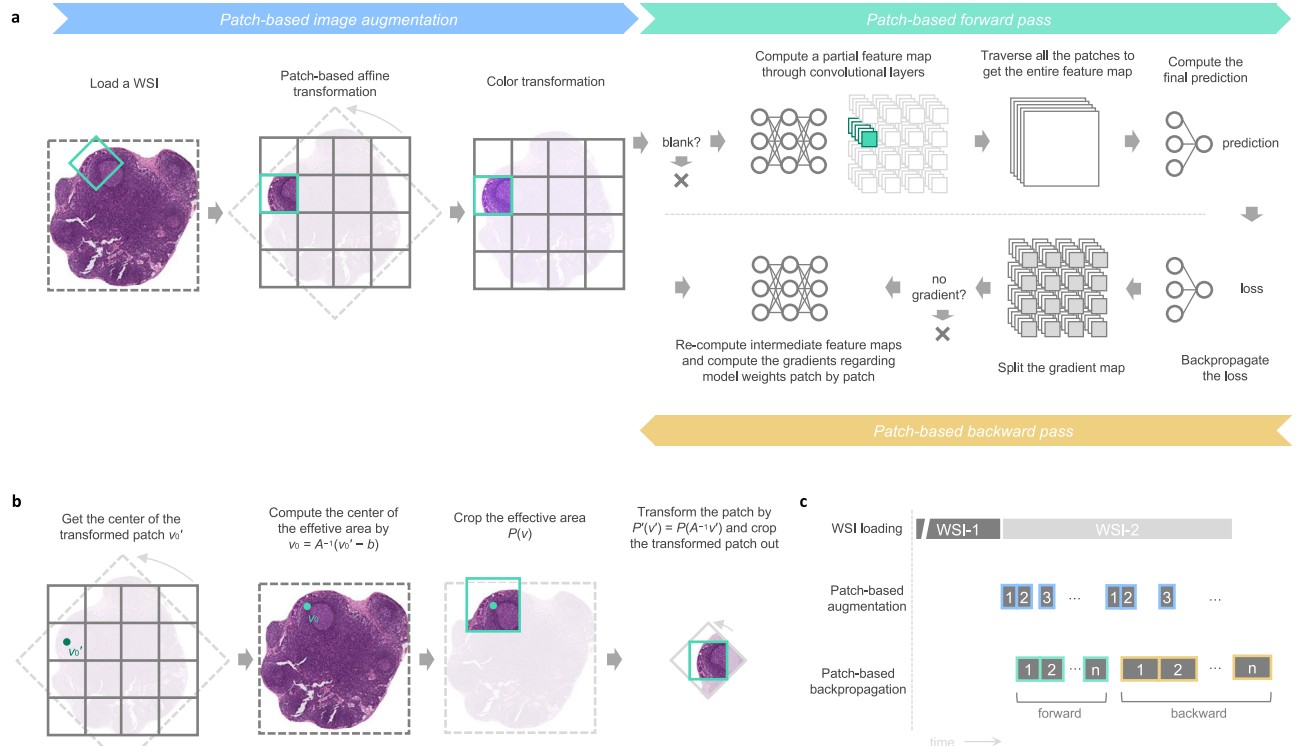

**Fig. 7 ESCNN as a weakly supervised method for training gigapixel images. a** Dataflow of ESCNN illustrates how patching mechanisms are embedded into image augmentation and the backpropagation algorithm (including forward pass and backward pass). **b** Procedure of the patch-based affine transformation. **c** Three threads run concurrently to fully utilize computing resources, including one for loading WSIs (Input/Output-intensive), one for patch-based image augmentation (bus-intensive), and one for patch-based backpropagation (GPU-intensive).

parameters (i.e., those with which the lowest validation loss was achieved) were saved. The software stack comprised CUDA 11.1 and cuDNN 7.6 for GPU acceleration, PyTorch 1.8.2 and Torchvision 0.9.2 for model construction and training, Open MPI 4.0.1 and Horovod 0.22.1 for multi-GPU training, OpenCV 4.1.1.26 and Pillow 8.2.0 for image processing, and OpenSlide 3.4.1 to decode WSIs.

The implementations of MIL and MIL-RNN were revised slightly according to an open-source implementation (https://github.com/aetherAI/whole-slide-cnn)[22] to support the same set of image augmentation processes and implement the modified ResNet50[26] used for training the ESCNN model. The patch size and the sequence length of the RNN were set at 224 × 224 and 10, respectively. The loss and optimizer were controlled, as were those for the ESCNN model. The CLAM model was trained using the default configuration of the official implementation (https://github.com/mahmoodlab/CLAM), which was published by the research group proposing the method[18]. Specifically, the patch size was set at 256 × 256, the feature extraction model was ResNet50 pre-trained with the ImageNet data set[48], the aggregation model was a single-attention-branch CLAM model, the loss for instance-level clustering was the Smooth SVM loss, and the loss for bag-level predictions was the cross-entropy. The Adam optimizer was employed to update the model with a learning rate of 2e−4.

**Evaluation of the clinical impact of the AI-assisted LN assessment workflow**. To evaluate the clinical impact of the present workflow, 80 slides were sampled from the archive of Linkou CGMH in 2020. Senior pathologist S.-C.H. verified the category of each slide with the assistance of IHC slides. Six attending pathologists (J.L., T.-Y.H., H.-C.C., K.-H.C., R.-C.W., Y.-J.L.) from Kaohsiung CGMH, Keelung CGMH, Chiayi CGMH, and Linkou CGMH were recruited. Each pathologist reviewed all 80 slides twice in both the AI-assisted (A) mode and the normal (N) mode. Half of the slides were randomly sampled to be reviewed first in A mode and then in N mode. The remaining 40 slides were reviewed first in N mode and then in A mode. To eliminate the carryover effect, the experiment was divided into two rounds separated by a 2-to-3-week washout period. In each round, 40 A-mode slides and 40 N-mode slides were reviewed. To minimize sampling bias, the order in which the slides were presented differed for each pathologist.

The experiment was designed to simulate a real-world clinical setting such that the actual impact of the proposed workflow on the pathologists' daily routine could be determined. The pathologists were required to calculate the number of positive and negative LNs on each slide and were obligated to review 80 slides within 2–3 days in each round. No strict time limit for slide review was imposed because such pressure might increase pathologists' tendency to rely on AI assistance.

The pathologists were instructed to prioritize diagnostic accuracy and were informed that the review time was recorded. A round was split into four phases. In each phase, 20 randomly sampled slides were reviewed. To prevent bias, the phases alternated between presentation in the ANNA manner and the NAAN order (equally distributed among the six pathologists). To ensure accurate time measurement, each phase was completed without interruption. In mode A, LNs were marked negative or positive, lesion areas were highlighted, and the total number of LNs was calculated in advance. By contrast, in mode N, only WSIs were displayed. To enable familiarization with the experimental procedure, the pathologists reviewed training images in both modes before the formal experiment began.

**Statistical analysis and evaluation metrics**. Regarding the laboratory experiments of model performance, we used the main test set consisting of 201 slides and 1156 LN images, the ground truths of which were reviewed by four pathologists. The LN detector was evaluated using the IoU metric (scikit-learn 0.22.1). The confidence intervals (CIs) of AUCs employed to score LN metastasis identification models, along with the two-sided P values, were calculated using the DeLong method[52] (pROC 1.18.0). To compare the performance of the models with that of the pathologists, the prediction scores of the models were binarized and evaluated for their sensitivities, specificities, positive predictive values (PPVs), negative predictive values (NPVs), and MCCs. The CIs of these metrics were retrieved using exact binomial confidence limits (epiR 2.0.35). The MCCs and their accompanying CIs were calculated using mltools 0.3.5 and the bootstrapping method ($n = 10,000$), respectively.

In the clinical experiments for testing the efficiency of the AI-assisted assessment workflow, the review time was modeled using linear mixed-effect models (lme4 1.1.27.1). In those models, the fixed effects were the mode (A or N), round, and category (normal, macrometastasis, micrometastasis, or ITCs). The random effects were the pathologist and the slide. Post hoc analyses were conducted to compare factors of interest (e.g., AI assistance) using Tukey's test (emmeans 1.6.3). Sensitivities and specificities were evaluated at slide level (positive: at least one positive LN) by using the exact binomial test (epiR 2.0.35 for CI computation and DTComPair 1.0.3 for comparison). CVs for quantifying the consistency of LN counts were calculated with unbiased estimates of variance (EnvStats 2.4.0). The two-sided P value for comparing the mean CVs of two groups was obtained through a paired t test. In addition, a transparent reporting of a multivariable prediction model for individual prognosis or diagnosis (TRIPOD) checklist is provided as Supplementary Table 2.

## Reporting summary

**Reporting summary**. Further information on research design is available in the Nature Research Reporting Summary linked to this article.

## Data availability

The raw experimental data are provided as Source Data, including pathologists' diagnoses and the model-predicted scores on the main test set (Fig. 2 and Table 1), the throughput and memory consumption of the training methods (Fig. 3), the review time and reported LN counts corresponding to each pathologist-slide obtained from the clinical experiment (Fig. 6). To protect patients' privacy, the slide data are not publicly available. Although, for reproducing the results in this study, researchers can request the corresponding authors, Chao-Yuan Yeh or Tse-Ching Chen, to access these slide data remotely through virtual private networking (VPN) with approval of the Institutional Ethics Committee of the Chang Gung Memorial Hospital (irb1@cgmh.org.tw). The requests will be processed in 10 business days. The ImageNet data set[48] used to pre-train models is publicly accessible at https://www.image-net.org.

## Code availability

The source code of this study can be downloaded from https://github.com/aetherAI/hms2 under the CC BY-NC-SA 4.0 license[53]. This includes a training pipeline seamlessly adaptable to other pathological cases, and a demo video that gives a brief overview of this study.

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

## Acknowledgements

This work was supported by grants from Taiwan's Ministry of Science and Technology (109–2320-B-182A-010-MY3 [S.-C.H.]), Ministry of Health and Welfare (MOHW111-TDU-B-221–014011 [S.-C.H.] and MOHW111-TDU-B-221-114009 [T.-C.C.]), and the Chang Gung Memorial Hospital (CMRPG5J0091 [S.-C.H.] and CMRPG3J0963 [T.-C.C.]). We thank Shuen-Tai Wang and Te-Min Chen of the National Center for High-Performance Computing for providing computing resources and technical assistance. This manuscript was edited by Wallace Academic Editing.

## Author contributions

S.-C.H., C.-Y.Y., and T.-C.C. planned the study. C.-C.C. and Y.-J.H. designed the ESCNN. C.-C.C., Y.-J.H., L.-W.T., A.-F.H., and S.-H.H. conducted the performance analysis and optimization of the ESCNN. C.-C.C. implemented the codes for model training and inference. T.-S.O. implemented the web-based platform for the clinical experiment. S.-C.H., C.-C.C., and M.-Y.C. designed the experiments. S.-C.H. and I.-C.L. collected the slides. S.-C.H., J.L., T.-Y.H., H.-C.C., and K.-H.C. performed the LN-level annotations for model training and testing. S.-C.H. made the final judgments of equivocal LNs with reference to IHC slides. J.L., T.-Y.H., H.-C.C., K.-H.C., R.-C.W., and Y.-J.L. participated in the clinical experiment. M.-Y.C. and T.-S.O. conducted the clinical experiment. C.-C.C. organized the experimental results and performed the quantitative analyses. S.-C.H. reviewed the results and performed the qualitative analyses. C.-Y.Y. and T.-C.C. critically reviewed and approved the manuscript. All authors contributed to the preparation of the manuscript.

## Competing interests

C.-Y.Y. is the founder, chairman, and chief executive officer of aetherAI. C.-C.C. is a data scientist of aetherAI. M.-Y.C. is a project manager of aetherAI. T.-S.O. is a backend engineer of aetherAI. The remaining authors declare no competing interests.
