## [Peer Review File · Nature Communications]

Reviewers' Comments:

Reviewer #1:

Remarks to the Author:

This manuscript describes the evaluation of a deep learning network to detect metastases in histological sections of gastric cancer lymph nodes. This is a clinically relevant question. There is a need to have clinically well-validated assays in this field. The technical innovation is also interesting and the authors compare their approach to appropriate benchmarks, including the CLAM model which has inspired a number of research studies.

In general, the authors propose a weakly supervised system which learns from slide labels only. This approach is pretty standard in the field right now, although the authors provide some modifications related to extraction of patches online during model training. Very few groups really use strongly supervised patch-based approaches anymore for large studies with 1000s of patients, because this is too labor intensive. This should be therefore down-toned in the introduction.

A nice aspect of this study is that the authors also evaluated the review time, which makes this paper attractive for a practical audience of surgical pathologists.

Another nice aspect is the definition of an actual threshold value which we do not see very often in these types of studies. However, it is not entirely clear if these thresholds were pre-defined before evaluation of the test set, or if they were cherry-picked post hoc.

The introduction contains a large bit of the results. Everything after "Our contributions and findings are summarized as follows." should be moved to other sections. Likewise, the results section contains an entire methods paragraph ("AI-assisted LN assessment workflow") which should be moved. Also, the discussion partly re-iterates the results which is not optimal.

The design of the study is concise, the text is written in a clear way and the figures have a high quality. Clearly, this is the work of professionals (i.e. a company), and not just individual academic researchers. This is of course nice, but related to one of my major concerns / questions: Which parts of this study represent a research algorithm, and which parts represent a commercial product? Have the authors marketed their solution or are they close to launching it as a product. This is currently not clear to me from the text and should be prominently acknowledged, i.e. in the abstract.

A strong limitation is that the source codes are not being made available. Of course, this is understandable if what is being tested here is a commercial product. In that case, however, a lightweight version of the core algorithm should be made available for academic evaluation. See, for example Campanella et al., Nature Medicine 2019, who published a basic version of their source codes and used the full version for their product.

Another limitation is that the relevant reporting guidelines do not seem to be included in this article, such as the MI-CLAIM guideline or TRIPOD-ML / any other relevant guidelines from the Equator network.

This might be picky, but "The primary challenge of direct WSI training is that excessive memory usage causes out-of-memory errors" is misleading. It is not the *error messages* which are problematic, but the fact that huge models just do not fit on the GPU VRAM. This sentence should be rephrased.

In general, the methods section is sometimes too long (sometimes it reiterates textbook knowledge), at other times it is not detailed enough (regarding the actual architecture of the model, for example). This could be optimized.

Table 1, how was the operating point found? Are these values for a predefined threshold, or a threshold which was optimized on the test set?

Reviewer #2:

Remarks to the Author:

Summary

In this paper, the authors developed an end-to-end weakly supervised method, termed enhanced streaming CNN (ESCNN), to facilitate the counting and diagnosis of lymph nodes (LNs) in gastric cancer. Unlike traditional patch-based deep learning approaches, the proposed method directly trains a CNN using giga-pixel images without lesion-level annotations. Based on large-scale datasets, the proposed system identified metastatic LNs with a slide-level AUC of 0.9936. In addition, the authors integrated the model into a pathological LN assessment workflow and conducted extensive clinical simulations, which demonstrated that the workflow significantly improved the sensitivity of identifying micrometastases and isolated tumor cells. The cross-site evaluation further validated that the algorithm's robustness.

Overall, this paper is well-written, well-organized, and a good exploration of applying streaming CNNs to reduce memory load and annotation requirements in digital pathology image analysis. Importantly, the authors developed sufficient experiments to demonstrate both the model's effectiveness and practicality. The proposed model consistently outperformed baseline methods in various designed tasks, especially for isolated tumor cells detection (from 67.95% to 96.15%). However, I still have several concerns related to the model architecture, experiment design, and evaluation. These are organized into major and minor concerns below with the related text from the manuscript quoted.

Scientific Premise:

Concern 1 (Major)

"The challenge of identifying metastatic carcinoma in LNs in gastric cancer is attributable to the high percentage of diffuse and mixed-type cancer, which accounts for more than 40% of cases. In diffuse and mixed-type gastric cancer, metastatic tumor cells may be poorly cohesive small clusters or individual cells. They may also resemble histiocytes or lymphocytes in appearance; that is, they have no well-adherent aggregates, glandular structures, and easily recognizable nuclear pleomorphism."

- The manuscript could benefit from better stating the motivation and premise for the work. As mentioned by the authors, many gastric cancer cases contain small clusters or individual tumor cells. It is worth noting that the proposed method improved the performance of detecting micrometastasis and isolated tumor cells (ITC) significantly. The authors could provide more details about the biological and diagnostic significance of isolated tumor cells and micrometastasis in gastric cancer.

- Previous research [1,2] has indicated that the clinical significance of detecting ITC in the lymph nodes of gastric cancer patients is controversial. Can the authors discuss more on the practical benefits of the proposed model's advantages in detecting small tumor regions?

Methodology:

Concern 2 (Minor)

"To reduce GPU memory consumption, the ESCNN follows the technique employed in streaming CNNs, involving the use of patching and gradient checkpointing. Both the forward and backward passes are divided into subtasks in the spatial dimensions. Each subtask handling a small region is sequentially executed to obtain a partial result, which is collected and then compiled with the others to obtain the full result. This divide-and-conquer technique saves memory space"

- Gradient checkpointing reduces memory usage by recomputing activations on the fly and storing them at strategic nodes of the computational graph. Streaming CNN accomplishes the same goal, but by splitting the problem into sub-problems that can be solved independently and provide intermediate equivalent results. However, streaming CNN is best suited to image classification or regression applications that use pooling/dilated convolutions and is not immediately applicable to

semantic segmentation or other network structures, which limits the potential application of the proposed method on other related image analysis tasks.

Concern 3 (Major)

"Instead of the reuse of intermediate feature maps in the typical backpropagation algorithm, patches of intermediate feature maps generated in the forward pass are released and recomputed in the backward pass. This process is called gradient checkpointing. By controlling the patch size such that each subtask is within the memory capacity of the GPU, this technique enables images of varying sizes to be processed on a GPU"

- The proposed method has the drawback of requiring the activation maps of the tiles to be recalculated during the backward pass, which can result in significant computational overhead. Aside from the time trade-off, architectures are less flexible since operations that require the entire activation map to be present, such as batch normalization, cannot be used.
- The runtime cost of the proposed method is unclear. The authors need to analyze algorithms by considering both time complexity and space complexity. When comparing the proposed method with baseline models, the authors should report the actual code running time and memory cost in detail.

Concern 4 (Major)

"Aside from the memory capacity issue, the training throughput decreased sublinearly as the image size increased, hindering the application of gradient checkpointing to demanding tasks requiring larger images (e.g., metastasis detection). This performance bottleneck was observed during image augmentation, which entails the sequential application of processing steps to an image to increase data variety. The extremely high memory footprint involved in transforming a high-resolution image into another in each processing step increases paging overhead and memory loading (i.e., thrashing) considerably. To resolve this problem, the ESCNN introduces patching to image augmentation such that the locality of the memory access pattern can be increased to maximize the efficiency of the memory system."

- In terms of model architecture design, this paper is conventional. The authors improve the ability of streaming CNN to handle image augmentation. However, the design motivation and the performance gain of the proposed image augmentation module are unclear. Why is this an important module to be added to the conventional streaming CNN? How large is the performance gain compared to the conventional streaming CNN? How large is the memory cost improvement of the proposed technique?
- The authors should consider performing ablation studies for the image augmentation module and also report the performance of the conventional streaming CNN.

Concern 5 (Minor)

"With the ESCNN training pipeline, we trained the metastasis identification model of the ResNet50 architecture with frozen batch normalization layers and global maximum pooling."

- What does "frozen batch normalization" mean? Has the ResNet been pre-trained on ImageNet? Since batch normalization breaks the local properties of chained convolutional and pooling layers, the authors of streaming CNN replaced batch normalization with LSUV initialization and weight decay. The authors should explain more details of using batch normalization layers in this paper.

Evaluation:

Concern 6 (Minor)

"To determine the effectiveness of the workflow, we designed a clinical experiment simulating a pathologist's routine LN assessment procedure. In the experiments, six pathologists reviewed 80 slides with and without AI assistance, and the review time and number of LNs positive and negative for metastatic carcinoma were recorded."

"To address false predictions, pathologists can edit contours, contour labels, or amend the final counts for correction. To assist pathologists with N-category assessment, a panel summarizing the numbers of positive and negative LNs of the current slide and study was employed"

- Additional information on timing would assist in assessing the significance of the results here. For example, how long did the program take to run? How much time did pathologists use to analyze or correct the prediction results from the model?

Concern 7 (Major)

“To investigate the impact of lesion sizes on the model performance, two subsets of the main test set were established. Each comprised all 861 negative LN images. One contained the 58 positive LN images demonstrating only micrometastasis (≥ 0.2 mm, < 2 mm), and the other contained the 28 positive LN images demonstrating only ITCs (< 0.2 mm). The model achieved AUCs of 0.9940 (0.9892–0.9988) and 0.9228 (0.8643–0.9814) on the micrometastasis and ITC test subsets, respectively. The results indicated that the ITC identification accuracy of the model remained to be improved.”

- The subset datasets for micrometastasis and ITC are highly unbalanced. In addition to AUC, it is better to further report results of other evaluation metrics, including F1 score and Matthews correlation coefficient.

- The proposed method significantly improved ITC detection performance. It would be interesting to have model interpretability analysis. For example, can the authors visualize the attention maps of the different methods to investigate the impact of the model's receptive field size on detecting small ROIs? Good results could further validate the advantages of the proposed approach.

Concern 8 (Major)

- Three baseline methods are included in this paper: slide-level MIL with max pooling, MIL-RNN, and attention MIL under the clustering constrained. However, there seem to be a lot of methods in the space of semi-supervised learning for digital pathology image analysis. Can the authors explain the rationale for why these are the best methods to compare against? Have these methods been identified as state-of-the-art in previous benchmarks?

- How were the hyper-parameters of baseline methods optimized? Has the same parameter search strategy been applied for both the proposed method and baseline methods?

Concern 9 (Major)

“The implementations of MIL and MIL-RNN were revised slightly according to an open-source implementation so that the common training parameters consist of those of the ESCNN model. The patch size and the sequence length of the RNN were set as 224×224 and 10, respectively. The CLAM model was trained using the default configuration of the official implementation published by its research group.”

- The model architecture of baseline methods is missing. What are the common training parameters? The authors should provide more details of implementing baseline methods.

References

1. Fukagawa, Takeo, et al. "The prognostic significance of isolated tumor cells in the lymph nodes of gastric cancer patients." *Gastric Cancer* 13.3 (2010): 191-196.
2. Horstmann, Olaf, et al. "Significance of isolated tumor cells in lymph nodes among gastric cancer patients." *Journal of cancer research and clinical oncology* 130.12 (2004): 733-740.

**Deep neural network trained on gigapixel images improves lymph node metastasis
detection in clinical settings**

Point-by-point response

Reviewer #1 (Remarks to the Author):

This manuscript describes the evaluation of a deep learning network to detect metastases in histological sections of gastric cancer lymph nodes. This is a clinically relevant question. There is a need to have clinically well-validated assays in this field. The technical innovation is also interesting and the authors compare their approach to appropriate benchmarks, including the CLAM model which has inspired a number of reserach studies.

Thanks for the review and valuable feedback. We hope the revised manuscript would better elaborate the proposed method and the clinical values of AI assistance.

In general, the authors propose a weakly supervised system which learns from slide labels only. This approach is pretty standard in the field right now, although the authors provide some modifications related to extraction of patches online during model training. Very few groups really uses strongly supervised patch-based approaches anymore for large studies with 1000s of patients, because this is too labor intensive. This should be therefore down-toned in the introduction.

We agree that adopting weakly supervised training is quite common in recent studies. Therefore, we have shortened the descriptions of the advantages of weak supervision over the strongly-supervised patch-based method in **Main** and **Discussion** (substantially in the first and the second paragraphs).

A nice aspect of this study is that the authors also evaluated the review time, which makes this paper attractive for a practical audience of surgical pathologists.

The reduction of review time is quite an attractive benefit brought from AI, especially in detecting LN metastasis, which accounts for certain workloads in a pathologist's daily practices.

Another nice aspect is the definition of an actual threshold value which we do not see very often in these types of studies. However, it is not entirely clear if these thresholds were pre-defined before evaluation of the test set, or if they were cherry-picked post hoc.

We defined the threshold by optimizing MCC on the main validation set. Please refer to the added description in the section **Results - ESCNN performance in metastasis identification**.

The introduction contains a large bit of the results. Everything after "Our contributions and findings are summarized as follows." should be moved to other sections. Likewise, the results section contains an entire methods paragraph ("AI-assisted LN assessment workflow") which should be moved. Also, the discussion partly re-iterates the results which is not optimal.

We have removed the paragraph starting with "Our contributions and findings are summarized as follows." since it is indeed not necessary to summarize the results right before the Results section. However, we decided to keep the section **AI-assisted LN assessment workflow** in **Results** as the first paragraph after the introduction section, considering readers may need a whole picture of the workflow before digging into the performance evaluation of each individual component. The **Discussion** section has been substantially shortened by removing redundant content in the first and second paragraphs.

The design of the study is concise, the text is written in a clear way and the figures have a high quality. Clearly, this is the work of professionals (i.e. a company), and not just individual academic researchers. This is of course nice, but related to one of my major concerns / questions: Which parts of this study represent a research algorithm, and which parts represent a commercial product? Have the authors marketed their solution or are they close to launching it as a product. This is currently not clear to me from the text and should be prominently acknowledged, i.e. in the abstract.

We are a research group consisting of professionals from academia and industry. Currently, only the aetherSlide digital pathology system (aetherAI, Taipei, Taiwan) for LN annotations is a product, which had been acknowledged in **Methods - Data preparation for model training and evaluation**. Most artifacts presented in this study, including ESCNN and the workflow, are research results instead of commercial products that have already been

launched or are to be launched in near future. (although we are working toward the commercialization of the research artifacts now.)

A strong limitation is that the source codes are not being made available. Of course, this is understandable if what is being tested here is a commercial product. In that case, however, a lightweight version of the core algorithm should be made available for academic evaluation. See, for example Campanella et al., Nature Medicine 2019, who published a basic version of their source codes and used the full version for their product.

We plan to release the source code on Github for readers to reproduce the results upon acceptance. The core part will be open-sourced, including the implementation of streaming CNN, the patch-based affine transformation algorithm, and the skipping mechanism. The remaining part will be runnable compiled codes. The code should be sent to you by the editor as an attached ZIP file if you would like to take a look.

Another limitation is that the relevant reporting guidelines do not seem to be included in this article, such as the MI-CLAIM guideline or TRIPOD-ML / any other relevant guidelines from the Equator network.

The completed “TRIPOD Checklist for Prediction Model Development and Validation” checklist has been included in **Supplementary Information**.

This might be picky, but "The primary challenge of direct WSI training is that excessive memory usage causes out-of-memory errors" is misleading. It is not the *error messages* which are problematic, but the fact that huge models just do not fit on the GPU VRAM. This sentence should be rephrased.

We have rephrased “out-of-memory errors” into “cannot fit in the limited memory capacity of a graphics processing unit (GPU)” in the second paragraph of **Methods - ESCNN for gastric LN metastasis identification**.

In general, the methods section is sometimes too long (sometimes it reiterates textbook knowledge), at other times it is not detailed enough (regarding the actual architecture of the model, for example). This could be optimized.

We have moved the entire section **Patch-based affine transformation algorithm** to **Supplementary Information** since it is quite straightforward with the illustration in **Fig. 7b**. As for your latter concern, we have extended the descriptions of how we modified ResNet50 in the last paragraph of **Methods - ESCNN for gastric LN metastasis identification**, and the detailed parameters (incl. architecture) of CLAM and MIL-RNN in the last paragraph of **Methods - Model training**.

Table 1, how was the operating point found? Are these values for a predefined threshold, or a threshold which was optimized on the test set?

The threshold for our model was determined by optimizing the MCC on the main validation set, as described in **Results - ESCNN performance in metastasis identification**.

Reviewer #2 (Remarks to the Author):

Summary

In this paper, the authors developed an end-to-end weakly supervised method, termed enhanced streaming CNN (ESCNN), to facilitate the counting and diagnosis of lymph nodes (LNs) in gastric cancer. Unlike traditional patch-based deep learning approaches, the proposed method directly trains a CNN using giga-pixel images without lesion-level annotations. Based on large-scale datasets, the proposed system identified metastatic LNs with a slide-level AUC of 0.9936. In addition, the authors integrated the model into a pathological LN assessment workflow and conducted extensive clinical simulations, which demonstrated that the workflow significantly improved the sensitivity of identifying micrometastases and isolated tumor cells. The cross-site evaluation further validated that the algorithm's robustness.

Overall, this paper is well-written, well-organized, and a good exploration of applying streaming CNNs to reduce memory load and annotation requirements in digital pathology image analysis. Importantly, the authors developed sufficient experiments to demonstrate both the model's effectiveness and practicality. The proposed model consistently outperformed baseline methods in various designed tasks, especially for isolated tumor cells detection (from 67.95% to 96.15%). However, I still have several concerns related to the model architecture, experiment design, and

evaluation. These are organized into major and minor concerns below with the related text from the manuscript quoted.

Thanks for your review and valuable feedback. We anticipate the revised manuscript would resolve all of your concerns and achieve higher quality.

Scientific Premise:

Concern 1 (Major)

“The challenge of identifying metastatic carcinoma in LNs in gastric cancer is attributable to the high percentage of diffuse and mixed-type cancer, which accounts for more than 40% of cases. In diffuse and mixed-type gastric cancer, metastatic tumor cells may be poorly cohesive small clusters or individual cells. They may also resemble histiocytes or lymphocytes in appearance; that is, they have no well-adherent aggregates, glandular structures, and easily recognizable nuclear pleomorphism.”

- The manuscript could benefit from better stating the motivation and premise for the work. As mentioned by the authors, many gastric cancer cases contain small clusters or individual tumor cells. It is worth noting that the proposed method improved the performance of detecting micrometastasis and isolated tumor cells (ITC) significantly. The authors could provide more details about the biological and diagnostic significance of isolated tumor cells and micrometastasis in gastric cancer.

• Previous research [1,2] has indicated that the clinical significance of detecting ITC in the lymph nodes of gastric cancer patients is controversial. Can the authors discuss more on the practical benefits of the proposed model's advantages in detecting small tumor regions?

Indeed, the prognostic impact of detecting micrometastases and ITCs is still under debate. Conservatively speaking, the clinical value of the sensitivity enhancement we can claim currently is to provide more information for clinical treatment (**Main**). Another interesting point we would like to discuss is the studies of the prognostic impact of micrometastasis and ITCs. This controversy may be due to the difficulty in collecting a sufficiently large number of ITC cases. Moreover, because of the challenges in detecting micrometastases and ITCs in clinical scenarios based on H&E-stained slides, the discrepant treatments applied to patients may act as a confounding factor that decreases the validity of studies. Employment of the AI-assisted workflow could hopefully resolve these issues. With the high detection sensitivity, more cases of interest would be screened out, and all patients would receive adequate treatment modalities. Given these, the real significance of micrometastases and ITCs would be revealed (**Discussion**).

Methodology:

Concern 2 (Minor)

“To reduce GPU memory consumption, the ESCNN follows the technique employed in streaming CNNs, involving the use of patching and gradient checkpointing. Both the forward and backward passes are divided into subtasks in the spatial dimensions. Each subtask handling a

small region is sequentially executed to obtain a partial result, which is collected and then compiled with the others to obtain the full result. This divide-and-conquer technique saves memory space”

- Gradient checkpointing reduces memory usage by recomputing activations on the fly and storing them at strategic nodes of the computational graph. Streaming CNN accomplishes the same goal, but by splitting the problem into sub-problems that can be solved independently and provide intermediate equivalent results. However, streaming CNN is best suited to image classification or regression applications that use pooling/dilated convolutions and is not immediately applicable to semantic segmentation or other network structures, which limits the potential application of the proposed method on other related image analysis tasks.

We would like to point out that typical strongly-supervised semantic segmentation and detection is not of interest in training using streaming CNN. Given annotations, the patching technique is available to resolve the memory issue in a simpler manner. Nevertheless, weakly-supervised semantic segmentation (WSSS) and object detection (WSOD) are relevant. Although the design space is huge, ESCNN can be immediately applicable to some architectures which use a first-staged classifier to generate pseudo labels via CAM (or other alternatives) and train a second-staged strongly-supervised downstream model using the pseudo labels. We can leverage ESCNN to produce pseudo labels for WSIs. This limitation and discussion are included in a new paragraph in the **Discussion** section.

Concern 3 (Major)

“Instead of the reuse of intermediate feature maps in the typical backpropagation algorithm, patches of intermediate feature maps generated in the forward pass are released and recomputed in the backward pass. This process is called gradient checkpointing. By controlling the patch size such that each subtask is within the memory capacity of the GPU, this technique enables images of varying sizes to be processed on a GPU”

- The proposed method has the drawback of requiring the activation maps of the tiles to be recalculated during the backward pass, which can result in significant computational overhead. Aside from the time trade-off, architectures are less flexible since operations that require the entire activation map to be present, such as batch normalization, cannot be used.

The overhead for recalculating feature maps is significant but is considered a cost-effective trade-off seeing the gigantic throughput leap brought by memory saving ($22.1\times$ speedup against the whole-slide training method). The inapplicability in running operations requiring global context is indeed a limitation of streaming CNN. A possible solution may be the halo exchange[3] mechanism that concurrently processes all patches so that inter-patch communication becomes possible, given a sufficient number of GPUs or TPUs. Although it is nearly impossible to get such a huge number of resources, the idea provides a good starting point to extend ESCNN. This discussion has been added in the **Discussion** section.

- The runtime cost of the proposed method is unclear. The authors need to analyze algorithms by considering both time complexity and space complexity. When comparing the proposed method with baseline models, the authors should report the actual code running time and memory cost in detail.

Thanks for the reminder that reporting the time and memory consumption is quite essential to this article. The key to making 20× WSI training feasible is enhancing the throughput of streaming CNN. A new section **Results - Throughput and memory consumption** and **Fig. 3** have been added to the manuscript, comparing the throughput (images/min.) and memory consumption (GB) of ESCNN with those of other end-to-end methods.

Concern 4 (Major)

“Aside from the memory capacity issue, the training throughput decreased sublinearly as the image size increased, hindering the application of gradient checkpointing to demanding tasks requiring larger images (e.g., metastasis detection). This performance bottleneck was observed during image augmentation, which entails the sequential application of processing steps to an image to increase data variety. The extremely high memory footprint involved in transforming a high-resolution image into another in each processing step increases paging overhead and memory loading (i.e., thrashing) considerably. To resolve this problem, the ESCNN introduces patching to image augmentation such that the locality of the memory access pattern can be increased to maximize the efficiency of the memory system.”

- In terms of model architecture design, this paper is conventional. The authors improve the ability of streaming CNN to handle image argumentation. However, the design motivation and the performance gain of the proposed image argumentation module are unclear. Why is this an important module to be added to the conventional streaming CNN? How large is the performance gain compared to the conventional streaming CNN? How large is the memory cost improvement of the proposed technique?

- The authors should consider performing ablation studies for the image augmentation module and also report the performance of the conventional streaming CNN.

The new section **Results - Throughput and memory consumption** should answer these concerns. On a $20\times$ model training, the training throughput of ESCNN was 0.912 images per min., $9.83\times$ faster than the 0.0928 images per min. of streaming CNN. The memory costs of streaming CNN and ESCNN are almost the same since they depend more on the patch size configuration.

Concern 5 (Minor)

“With the ESCNN training pipeline, we trained the metastasis identification model of the ResNet50 architecture with frozen batch normalization layers and global maximum pooling.”

- What does “frozen batch normalization” mean? Has the ResNet been pre-trained on ImageNet? Since batch normalization breaks the local properties of chained convolutional and pooling layers, the authors of streaming CNN replaced batch normalization with LSUV initialization and weight decay. The authors should explain more details of using batch normalization layers in this paper.

Yes, the ResNet is pre-trained by the ImageNet dataset (**Methods - Model training**). The term “frozen” means to keep a batch normalization layer always in the evaluation mode. We have added more detailed descriptions of it in **Methods -ESCNN for gastric LN metastasis identification**.

Evaluation:

Concern 6 (Minor)

“To determine the effectiveness of the workflow, we designed a clinical experiment simulating a pathologist’s routine LN assessment procedure. In the experiments, six pathologists reviewed 80 slides with and without AI assistance, and the review time and number of LNs positive and negative for metastatic carcinoma were recorded.”

“To address false predictions, pathologists can edit contours, contour labels, or amend the final counts for correction. To assist pathologists with N-category assessment, a panel summarizing the numbers of positive and negative LNs of the current slide and study was employed”

- Additional information on timing would assist in assessing the significance of the results here. For example, how long did the program take to run? How much time did pathologists use to analyze or correct the prediction results from the model?

Doing a breakdown on review time is actually a challenging task. Different pathologists own different habits to complete their tasks. Some prefer editing contours, while others prefer directly amending the LN counts. Some are proficient in multitasking—editing a contour and checking another LN in the meanwhile. Even though we have program traces, the actual motivation behind one event is still unknown.

Besides, program run time (about 5 min.) is excluded from the review time reported here. The model inference is triggered immediately after importing a WSI. This WSI is ready for pathologists verification after the inference is done. Therefore, no waiting time for loading the AI inference is needed for pathologists.

Concern 7 (Major)

“To investigate the impact of lesion sizes on the model performance, two subsets of the main test set were established. Each comprised all 861 negative LN images. One contained the 58 positive LN images demonstrating only micrometastasis (≥ 0.2 mm, < 2 mm), and the other contained the 28 positive LN images demonstrating only ITCs (< 0.2 mm). The model achieved AUCs of 0.9940 (0.9892–0.9988) and 0.9228 (0.8643–0.9814) on the micrometastasis and ITC test subsets, respectively. The results indicated that the ITC identification accuracy of the model remained to be improved.”

- The subset datasets for micrometastasis and ITC are highly unbalanced. In addition to AUC, it is better to further report results of other evaluation metrics, including F1 score and Matthews correlation coefficient.

Table 1 is extended to include more evaluation metrics other than AUCs on the two subsets, including confusion matrices, sensitivities, specificities, PPVs, NPVs, and MCCs. F1 scoring has been criticized for having a prior assumption that precision and recall are equally important and less truthful in some situations compared to MCC[4]. Therefore, we did not report F1 scores in the manuscript.

- The proposed method significantly improved ITC detection performance. It would be interesting to have model interpretability analysis. For example, can the authors visualize the attention maps of the different methods to investigate the impact of the model's receptive field size on detecting small ROIs? Good results could further validate the advantages of the proposed approach.

The added **Fig. 4** visualizes the localization of different methods, including the 5× ESCNN model, CLAM, and MIL. Some descriptions have been added in **Results - Lesion highlights and qualitative analysis**. The CAM result of our model demonstrated a higher coverage of macro- and micro-metastasis lesions, and possessed the ability to localize ITCs.

Concern 8 (Major)

- Three baseline methods are included in this paper: slide-level MIL with max pooling, MIL-RNN, and attention MIL under the clustering constrained. However, there seem to be a lot of methods in the space of semi-supervised learning for digital pathology image analysis. Can the authors explain the rationale for why these are the best methods to compare against? Have these methods been identified as state-of-the-art in previous benchmarks?

It is hard to identify state-of-the-art methods with the best performance because there is no benchmarking dataset for weakly-supervised learning on digital pathology currently. Therefore, we chose the baseline methods from three criteria: superiority, reproducibility, and peer review. First, a candidate method should be compared with at least one alternative and claim a superior performance in its paper. Second, the source code of a candidate method should be accessible to ensure reproducibility. This is important due to the difficulty in replicating exactly

the same implementation through the description in a paper; e.g., a hyper-parameter setting was seldom exhaustively listed in a paper. Third, a candidate method should be peer-reviewed to ensure the validity of its experimental results. MIL, MIL-RNN, and CLAM as well as streaming CNN and the whole-slide training method meet these 3 criteria and thus were chosen as the baseline methods. Other nice papers were added in the **References** section, including MR-MIL[5] (no source code) and Zoom-in Network[6] (no source code; not peer-reviewed yet).

- How were the hyper-parameters of baseline methods optimized? Has the same parameter search strategy been applied for both the proposed method and baseline methods?

We did not tune the hyper-parameters neither of our method nor the baseline methods. Training such a thousands-of-slide model using any weakly supervised method is both time- and power-consuming, taking weeks to train a model even with 32 V100 GPUs. To prevent resource spending, we directly ran the official implementations of baseline methods using the default hyper-parameters, which were considered optimal by the authors. By contrast, our method does not introduce additional hyper-parameters, so it does not require tuning. The hyper-parameter setting of baseline methods has been listed in the **Methods - Model training** section in detail.

Concern 9 (Major)

“The implementations of MIL and MIL-RNN were revised slightly according to an open-source implementation so that the common training parameters consist of those of the ESCNN model. The patch size and the sequence length of the RNN were set as 224×224 and 10, respectively.

The CLAM model was trained using the default configuration of the official implementation published by its research group.”

- The model architecture of baseline methods is missing. What are the common training parameters? The authors should provide more details of implementing baseline methods.

Please see the last paragraph of **Methods - Model training** for the model architecture and more details of the hyper-parameters of the baseline methods.

References

1. Fukagawa, Takeo, et al. "The prognostic significance of isolated tumor cells in the lymph nodes of gastric cancer patients." *Gastric Cancer* 13.3 (2010): 191-196.
2. Horstmann, Olaf, et al. "Significance of isolated tumor cells in lymph nodes among gastric cancer patients." *Journal of cancer research and clinical oncology* 130.12 (2004): 733-740.
3. Shazeer, N. *et al.* Mesh-tensorflow: Deep learning for supercomputers. *Advances in neural information processing systems* **31**, (2018).
4. Chicco, D., Tötsch, N. & Jurman, G. The Matthews correlation coefficient (MCC) is more reliable than balanced accuracy, bookmaker informedness, and markedness in two-class confusion matrix evaluation. *BioData mining* **14**, 1–22 (2021).
5. Li, J. *et al.* A multi-resolution model for histopathology image classification and localization with multiple instance learning. *Computers in Biology and Medicine* **131**, 104253 (2021).
6. Kong, F. & Henao, R. Efficient Classification of Very Large Images with Tiny Objects. *arXiv preprint arXiv:2106.02694* (2021).

Reviewers' Comments:

Reviewer #1:

Remarks to the Author:

The authors have addressed all of the comments which I raised. Only the response to the source code is not satisfactory. I recommend that the paper is only accepted if the source codes are made publicly available on Github, as is standard in the field.

Reviewer #2:

Remarks to the Author:

I thank the authors for their point-by-point response. My major concerns have been addressed.

Concern 1: Authors have added the significance of detecting micrometastases and ITCs into both the background and discussion sections.

Concern 2: Authors have improved the discussion on model limitations and future work.

Concerns 3 and 4: Time and memory consumption have been reported in detail. Results have demonstrated that ESCNN achieved significant improvement compared to conventional and whole-slide training methods.

Concern 6: As the model inference is triggered immediately after importing a WSI, the concern on model inference time can be ignored.

Concern 7: Table 1 and visualization figures have been updated as suggested.

Concerns 5, 8, 9: Details of the model architecture and hyper-parameters of the baseline methods are reported in the revision. One minor concern is that the authors did not tune the hyper-parameters, which reduces the confidence in demonstrating the model design advantages. This is understandable as the hyper-parameter analysis is time- and power-consuming.

Recommendation:

I recommend that this paper be accepted.

**Deep neural network trained on gigapixel images improves lymph node metastasis
detection in clinical settings**

Point-by-point response

Reviewer #1 (Remarks to the Author):

The authors have addressed all of the comments which I raised. Only the response to the source code is not satisfactory. I recommend that the paper is only accepted if the source codes are made publicly available on Github, as is standard in the field.

Thanks for your valuable comments that improved the quality of this article. The source code is now publicly available on <https://github.com/aetherAI/hms2> . By following the procedure described in README.md, it should be straightforward to train and evaluate a model using the CAMELYON16 data set (or your own data sets if applicable). If that is not the case, please feel free to contact us.

Reviewer #2 (Remarks to the Author):

I thank the authors for their point-by-point response. My major concerns have been addressed.

We appreciate your review and recommendation, and feel delighted to resolve all your major concerns. The article has been improved considerably given your suggestions, especially

the reporting of throughput and memory consumption and the visualization comparisons that highlight the benefits of ESCNN.

Concern 1: Authors have added the significance of detecting micrometastases and ITCs into both the background and discussion sections.

Concern 2: Authors have improved the discussion on model limitations and future work.

Concerns 3 and 4: Time and memory consumption have been reported in detail. Results have demonstrated that ESCNN achieved significant improvement compared to conventional and whole-slide training methods.

Concern 6: As the model inference is triggered immediately after importing a WSI, the concern on model inference time can be ignored.

Concern 7: Table 1 and visualization figures have been updated as suggested.

Concerns 5, 8, 9: Details of the model architecture and hyper-parameters of the baseline methods are reported in the revision. One minor concern is that the authors did not tune the hyper-parameters, which reduces the confidence in demonstrating the model design advantages. This is understandable as the hyper-parameter analysis is time- and power-consuming.

Thanks for your understanding of the lack of hyper-parameter tuning. It is indeed a challenge to tune hyper-parameters given the time-consuming nature of developing a digital pathology AI. Therefore in this study, we prefer those designs robust and insensitive to hyper-parameter tuning; e.g., ResNet (which is seldom trapped in saddle points) and AdamW (instead of SGD which requires sophisticated learning rate scheduling) to ensure the quality of trained models given only a limited number of trials.

Recommendation:

I recommend that this paper be accepted.